# Decoding temporal interpretation of the morphogen Bicoid in the early *Drosophila* embryo

Anqi Huang[1†], Christopher Amourda[1†], Shaobo Zhang[1], Nicholas S Tolwinski[2,3], Timothy E Saunders[1,3,4*]

[1]Mechanobiology Institute, National University of Singapore, Singapore, Singapore; [2]Division of Science, Yale-NUS College, Singapore, Singapore; [3]Department of Biological Sciences, National University of Singapore, Singapore, Singapore; [4]Institute for Molecular and Cell Biology, Agency for Science Technology and Research, Singapore, Singapore

**Abstract** Morphogen gradients provide essential spatial information during development. Not only the local concentration but also duration of morphogen exposure is critical for correct cell fate decisions. Yet, how and when cells temporally integrate signals from a morphogen remains unclear. Here, we use optogenetic manipulation to switch off Bicoid-dependent transcription in the early *Drosophila* embryo with high temporal resolution, allowing time-specific and reversible manipulation of morphogen signalling. We find that Bicoid transcriptional activity is dispensable for embryonic viability in the first hour after fertilization, but persistently required throughout the rest of the blastoderm stage. Short interruptions of Bicoid activity alter the most anterior cell fate decisions, while prolonged inactivation expands patterning defects from anterior to posterior. Such anterior susceptibility correlates with high reliance of anterior gap gene expression on Bicoid. Therefore, cell fates exposed to higher Bicoid concentration require input for longer duration, demonstrating a previously unknown aspect of Bicoid decoding.

*For correspondence: dbsste@nus.edu.sg

†These authors contributed equally to this work

Competing interests: The authors declare that no competing interests exist.

## Introduction

Morphogens are molecules distributed in spatial gradients that provide essential positional information in the process of development (*Turing, 1990*; *Wolpert, 1969*). By activating differential gene expression in a concentration-dependent manner, morphogens instruct the cells to adopt proper cell fates according to their positions in the developing embryos or tissues (*Gurdon and Bourillot, 2001*; *Neumann and Cohen, 1997*). The impact of a morphogen gradient on a developing system depends on two characteristics: first, its information capacity in terms of how many distinct cell types it has an effect on; second, its transferring precision – in essence, how reproducible cell fates are in different individuals at given positions. Each of these characteristics depends not only on the local concentration of morphogen molecules that the cells interpret, but also temporal components of such interpretation.

The temporal pattern of morphogen interpretation has been demonstrated in several vertebrate developing systems. Harfe et al. first proposed that the length of time of morphogen signalling is critical for correct cell fate specification. They found that during mouse limb development, cells exposed to Sonic Hedgehog (Shh) morphogen for longer time develop into digits of more posterior identity (*Harfe et al., 2004*). Similarly, in chick neural tube formation the duration of Shh activity is translated into different cell types along the dorso-ventral axis (*Dessaud et al., 2007*). In comparison, in fish dorso-ventral patterning it is not the duration but the timing of BMP signaling that is

important for correct cell fate determination (*Tucker et al., 2008*). While in some cases the temporal integration of morphogen signaling is carried out by genetic feedback loops (*Dessaud et al., 2010*), in many other systems the underlying mechanism has not been unveiled.

As the first protein identified to function as a morphogen, Bicoid (Bcd) patterns the cells along the antero-posterior axis in the early embryo of the fruitfly, *Drosophila melanogaster* (*Driever et al., 1989*). Bcd is maternally deposited and localized at the anterior pole of the embryo in the form of mRNA (*Frohnhöfer and Nüsslein-Volhard, 1986*). This localized mRNA is further translated upon fertilization, forming a protein gradient with exponential decay in concentration along the antero-posterior axis. Fundamentally acting as a transcription factor, Bcd activates different genes, with the more anteriorly expressed genes having lower Bcd binding affinity (*Struhl et al., 1989*). Quantitative studies have shown that the Bcd gradient is highly dynamic, with nuclear Bcd concentration constantly changing throughout the first 13 rounds of syncytial cell divisions (*Little et al., 2011*). It has been suggested that precise positional information endowed by Bcd gradient at one single time point suffices to distinguish neighbor cell identities (*Gregor et al., 2007a*). However, recent results have motivated intense debate over what is the exact time window when the information carried by Bcd is interpreted (*Bergmann et al., 2007*; *Gregor et al., 2007b*; *Liu et al., 2013*; *Lucchetta et al., 2005*).

To address this question, fast and reversible temporal manipulation of Bcd activity is required. In this study, we have developed an optogenetic tool to control Bcd-dependent transcription in the early *Drosophila* embryo with high temporal resolution. Using this tool, we provide a detailed dissection of the temporal components of Bcd decoding in vivo. The results unveil an unexpected temporal pattern of Bcd action – cell fates determined by higher Bcd concentration require Bcd for longer duration, while cells experiencing lower Bcd dosage commit to correct fates at earlier developmental stages. Further combining temporal perturbation with quantitative analyses, we find that the differential expression kinetics of target genes can partly explain the mechanism underlying such temporal interpretation.

## Results

### Precise temporal control of Bcd-dependent transcription via optogenetic manipulation

We built a light-responsive construct to switch off Bcd-dependent transcription by fusing the optogenetic cassette cryptochrome 2 (CRY2) together with mCherry (mCh) to the N-terminus of Bcd (*Kennedy et al., 2010*) (*Figure 1A*). We used standard P-element transformation to insert this CRY2::mCh::Bcd construct at different genomic loci generating diverse fly lines expressing this fusion protein at various levels under the control of the endogenous bcd regulatory sequences.

We find that in the dark, CRY2::mCh::Bcd transgenics rescues the absence of endogenous Bcd. The expression patterns of four gap genes – hunchback (hb), giant (gt), krüppel (kr) and knirps (kni) – are reminiscent of wild-type embryos when the light-responsive Bcd construct is expressed in an otherwise *bcd* mutant background (*Figure 1B–E*). These embryos hatch as healthy larvae (*Figure 1H*). In comparison, embryos illuminated with blue light during the first 2.5 hr after egg deposition fail to develop the head and thorax structures (*Figure 1H'*), resembling that of the *bcd* knockout phenotype (*Figure 1H''*). Illuminated embryos show strong repression of the anterior, Bcd-dependent, expression of hb, gt and kni, while their posterior, Bcd-independent counterparts remain intact (*Figure 1B'–D'*). Furthermore, the Kr band shifts anteriorly compared to the pattern observed in the dark (*Figure 1E–E'*, arrowheads). Noticeably, while *bcd* null embryos are characterized by expression of ectopic hb and gt stripes in the anterior regions (*Figure 1B'' and C''*, arrowheads) and a duplicated telson at the anterior end (*Figure 1H''*, arrowhead), these features do not appear in illuminated embryos. Such phenotypic differences can be readily explained by the different distribution of Caudal (Cad) proteins in two different ways of Bcd perturbation. In wild-type embryos, Cad is expressed in a broad posterior domain in early blastoderm stages as a result of translational repression by Bcd and functions in posterior patterning by activating abdominal gap genes (*Schulz and Tautz, 1995*). While Cad is uniformly expressed in *bcd* null embryos due to a lack of anterior repression (*Figure 1G''*), we saw no shift in Cad gradient in illuminated embryos when compared to those in the dark (*Figure 1G,G' and K*). It suggests that light-induced conformational changes did not alter

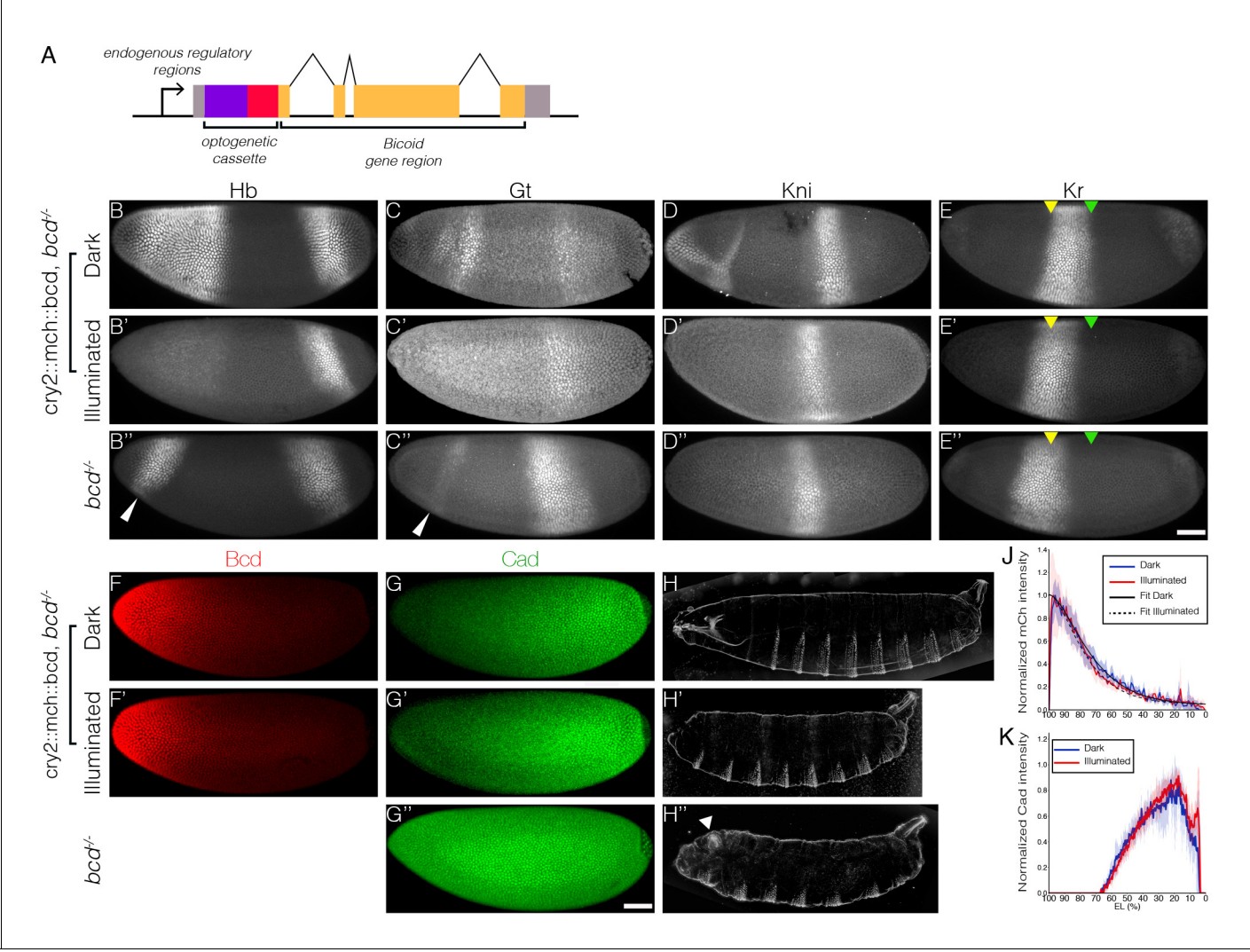

**Figure 1.** Optogenetic tool to manipulate Bcd-dependent transcription activity. (**A**) Schematic illustration of CRY2::mCh::Bcd construct; CRY2 optogenetic cassette tagged with mCherry fluorescent protein is fused to the N-terminal of Bcd coding sequence; expression of the construct is under the regulation of the endogenous *bcd* regulatory sequence. (**B–E**) Embryos maternally expressing *cry2::mch::bcd* having developed in the dark (**B–E**) or light (**B'–E'**) as compared to *bcd^-/-* embryos (**B''–E''**) fixed at the end of the blastoderm stage and stained for Hb (**B**), Gt (**C**), Kni (**D**) and Kr (**E**). Yellow and green arrowheads indicate, respectively, the anterior and posterior Kr boundaries in the dark. White arrowheads point to ectopic posterior Hb and Gt expression in the anterior. (**F and G**) Embryos in the dark (**F and G**) or light (**F' and G'**) compared to *bcd^-/-* (**G''**) stained for mCh (**F**) and Cad (**G**) at early n.c. 14. (**H**) Cuticle patterns of embryos with maternally loaded *cry2::mch::bcd* having developed in the dark (**H**) or light (**H'**) in the first 2.5 hr AEL as compared to *bcd^-/-* embryos (**H''**). White arrowhead indicates duplicated telson. (**J and K**) Average nuclear intensity of CRY2::mCh::Bcd (**J**) or Cad (**K**) normalized to peak value is plotted vs. AP position (% EL) in cry2::mch::bcd, *bcd^-/-* embryos having developed in dark (blue curve) and light (red curve). (**J**) Data were fitted to an exponential curve shown by smooth lines with length scales around 80 µm. Shaded error bars are across all nuclei of all embryos at a given position. (**B–K**) n = 5–7 embryos per condition. Scale bar, 50 µm.

The following figure supplement is available for figure 1:

**Figure supplement 1.** Caudal expression in embryos of different bcd dosage.

the translation inhibitory capability of Bcd in the cytoplasm (in comparison, Cad boundary shifts anteriorly with decreasing bcd dosages, *Figure 1—figure supplement 1*). We further quantified the Bcd gradient under dark and illuminated conditions (*Figure 1F–F'*) and observed no significant change in Bcd profile (*Figure 1J*), showing that the spatial distribution of Bcd molecules is not affected by our manipulation.

Next, we tested whether CRY2::mCh::Bcd under illumination inhibits transcription in a dominant negative manner. We selected two fly lines expressing CRY2::mCh::Bcd at comparatively lower and higher levels in a bcd wild-type background, denoted by CRY2$^{low}$ and CRY2$^{high}$, respectively (*Figure 2—figure supplement 1*) and utilized the MS2-MCP system to visualize the transcriptional activity of the Bcd target gene hb as shown by the GFP-tagged MS2 coat protein (MCP::GFP) (*Garcia et al., 2013*; *Lucas et al., 2013*). Compared to wild-type embryos - where transcriptional activity of hb in the Bcd-dependent anterior domain persists throughout the interphase of nuclear cycle (n.c.) 13 (7.1 ± 3.2 min; *Figure 2A and B*, *Video 1*) - CRY2$^{low}$ embryos show significantly reduced transcriptional persistence when illuminated with a 488 nm laser (3.6 ± 1.1 min; *Figure 2A and C*). Such an inhibitory effect becomes more prominent in CRY2$^{high}$ embryos (3.0 ± 0.5 min, *Figure 2A and D*). The transcription of the posterior hb domain is Bcd-independent and, reassuringly, we do not observe any inhibitory effect in this domain (4.3 ± 2.0 min, 5.2 ± 3.3 min, and 4.2 ± 2.0 min in control, CRY2$^{low}$, and CRY2$^{high}$, respectively; *Figure 2E*). Further, the inhibitory effect for the same genetic background and illumination conditions is uniform along the AP axis (*Figure 2—figure supplement 2*). As parallel evidence, ChIP-qPCR experiments show comparable CRY2::mCh::Bcd binding in dark or light to native Bcd binding sites (*Figure 2F*), with no noticeable unspecific DNA binding caused by light-induced conformational changes (*Figure 2G*).

To test the rate of light-dependent loss and recovery of transcription, we generated the MCP::mCh fly line to image the hb transcription activities throughout the course of illumination and dark recovery (*Figure 2—figure supplement 3A*). First, we validated that in a wild-type bcd background MCP::mCh shows the same reporter kinetics as MCP::GFP regarding the density of transcriptionally active nuclei as well as transcription persistence from n.c. 12 to 14 (*Figure 2—figure supplement 3B–E*). Further, under dark conditions embryos expressing maternally loaded CRY2::mCh::Bcd and MCP::mCh show the same hb transcription dynamics as that of wild-type embryos (*Figure 2—figure supplement 4A and A'*). When these embryos are subjected to short illumination (1 min) two minutes after the onset of hb transcription in n.c. 13, we observed an abrupt diminishing number of active nuclei over a period of 4 min, and a complete cessation of fluorescent signals 4 min after illumination (*Figure 2—figure supplement 4B and B'*). Given that the MS2-yellow reporter gene is 6.4 kb in length and the measured PolII elongation rate is 1.54 kb/min (*Garcia et al., 2013*), we attribute the remnant reporter signals within these 4 min to the elongating transcripts initiated before illumination. When the embryos were illuminated during the 12th division, no transcription activity was detected until 5 min after dark recovery in n.c.13, with a full recovery in n.c.14 (*Figure 2—figure supplement 4C and C'*). These observations suggest that the optogenetic manipulation shows fast kinetics with immediate inhibition of transcription upon illumination and re-initiation of transcription 5 min after returning to dark condition.

We then explored the effect of tuning the inhibition of Bcd-dependent transcription by varying the blue-light illumination intensity. We illuminated embryos with blue light at different powers (0.04, 0.4 and 4 mW) during the hour prior to gastrulation. We used the Even-skipped (Eve) pattern, where Eve stripes 1 and 2 are Bcd-dependent and more posterior stripes are Bcd-independent as a readout of Bcd activity (*Frasch and Levine, 1987*; *Stanojevic et al., 1991*). In a WT background, low power illumination (0.04 mW) causes a moderate reduction and a slight anterior shift of the Eve stripes 1 and 2 expression (*Figure 2H and I*). Increasing light power further decreases the expression level of the first two Eve stripes (*Figure 2J and K*). When expressed in a *bcd* null background, CRY2::mCh::Bcd completely inhibits the expression of the anterior two Eve stripes, even at the lowest light power tested (*Figure 2L–O*).

## Persistent Bcd transcription activity is indispensable to embryonic viability

Bcd confers robust cell fate decisions by activating hierarchical segmentation gene networks (*Jaeger et al., 2004*; *Kraut and Levine, 1991*; *Manu et al., 2009a*). Previous studies have shown that only at early n.c. 14 is the absolute Bcd concentration interpreted as positional information. In late n.c. 14, on the contrary, expression boundaries of downstream genes are subjected to cross-regulation and the Bcd gradient is believed to no longer be important (*Liu et al., 2013*). To determine whether Bcd-dependent transcription in late n.c. 14 is actually irrelevant for cell fate determination, we collected embryos laid by females expressing CRY2::mCh::Bcd in a *bcd* null background (CRY2$^{high}$, Bcd$^{null}$; in all following experiments unless otherwise stated) and illuminated them before

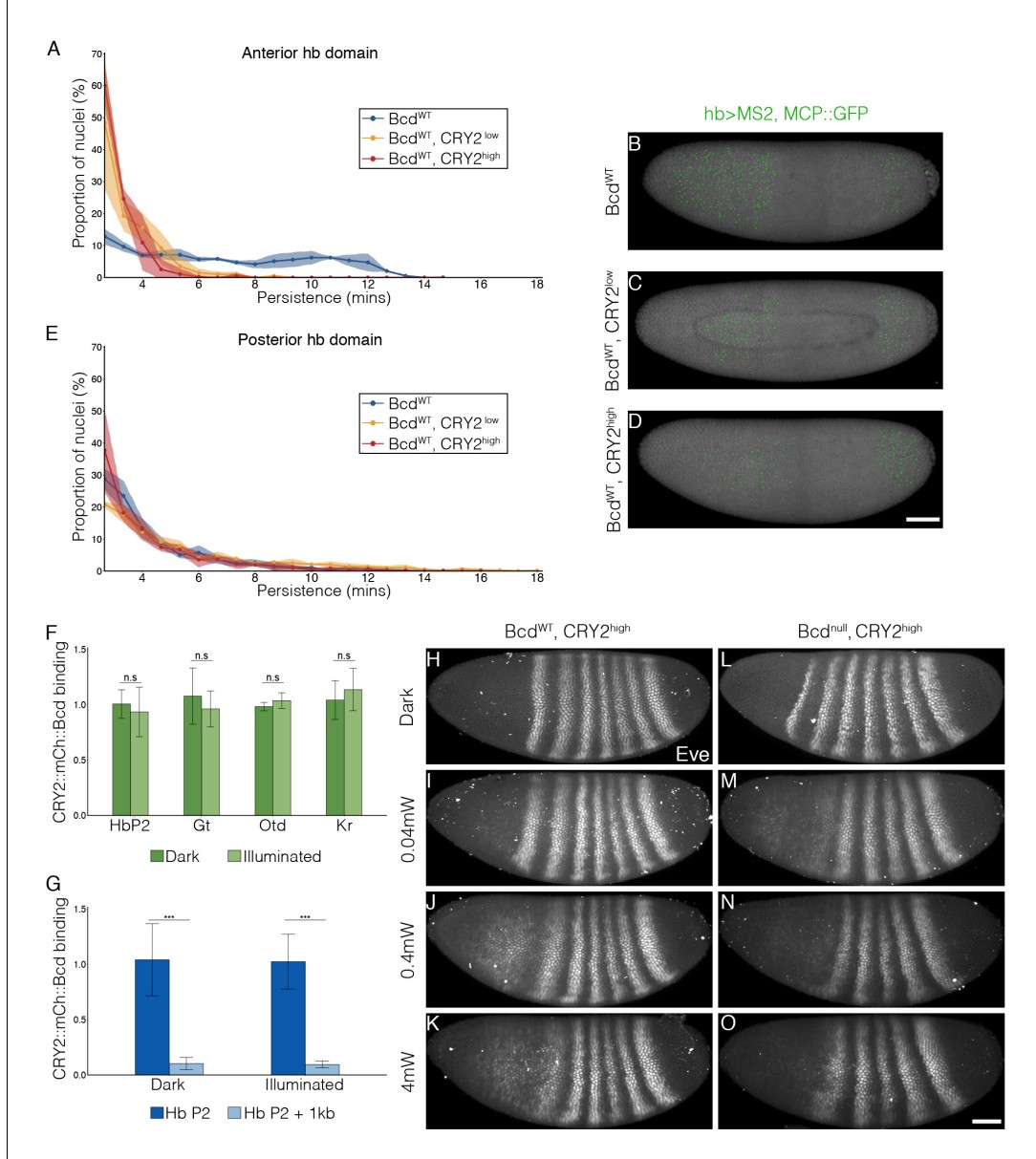

**Figure 2.** Inhibitory effect on Bcd-dependent transcription is gene-dosage and light-power dependent. (**A and E**) Proportion of the nuclei positive for *hb* transcription plotted vs. the persistence of transcription activity during n.c. 13 in control (blue), CRY2[low] (orange) and CRY2[high] (red) embryos of *hb* anterior (**A**) and posterior (**E**) domain. n = 3 embryos per genotype. (**B–D**) Snapshots of embryos expressing hb>MS2, MCP::GFP in n.c.13. Embryos are maternally loaded with only endogenous *bcd* (**B**), or together with *cry2::mch::bcd* at low(**C**) or high (**D**) level. MCP::GFP signals are tracked and marked with green dots. (**F and G**) Chromatin immunoprecipitation quantitative PCR (ChIP-qPCR) characterization of CRY2::mCh::Bcd binding to Hb (P2 enhancer), Gt, Otd and Kr in the light for 45 mins (4 mW) as compared to embryos aged in the dark (**F**) and to 1 kb downstream of the Hb P2 enhancer in the dark and light conditions (**G**). Data are presented as Mean ± SD, *p<0.001. (**H–O**) Eve staining in embryos maternally loaded with cry2::mch::bcd in bcd[WT] (**H–K**) or *bcd* null (**L–O**) background; embryos have developed in dark (**H and L**) or illuminated with blue light at 0.04 mW (**I and M**), 0.4 mW (**J and N**) or 4 mW (**K and O**). n = 5–7 embryos per condition. Scale bar, 50 μm.

The following figure supplements are available for figure 2:

**Figure supplement 1.** Expression profiles in CRY2[low] and CRY2[high] embryos.

**Figure supplement 2.** Effect of illumination on hb mRNA production along the AP axis.

**Figure supplement 3.** MCP::mCh shows the same reporter kinetics as MCP::GFP.

*Figure 2 continued on next page*

*Figure 2 continued*

**Figure supplement 4.** Rate of light-induced loss and recovery of Bcd-dependent transcription.

the initiation of gastrulation for time windows ranging from 10 to 60 min (*Figure 3A*). Surprisingly, 10 min of illumination at the end of n.c. 14 is sufficient to induce embryonic lethality. Cuticle preparation of these non-hatched embryos shows defective mouthparts, in particular the pharynx (*Figure 3C and D*, arrows). In addition, the structures between mouth hooks and cephalo-pharyngeal plates are missing (*Figure 3D*, arrowhead). Thirty minutes of illumination causes further deterioration in mouth development, with minimal mouth skeleton visible (*Figure 3E*). Extending the illumination window depletes all mouthparts (*Figure 3F and G*), with the most severe phenotypes observed when illumination encompasses both n.c. 13 and n.c. 14. Under this condition, all thorax segments are lost, with only abdominal denticle belts remaining (*Figure 3G*).

Transcripts of Bcd target genes have been detected using in situ hybridization at time points as early as n.c. 7 (*Ali-Murthy and Kornberg, 2016*). We tested whether such early Bcd-dependent transcription was essential for embryonic viability by illuminating embryos during early developmental stages (*i.e.* before n.c. 14) and reverting them later to dark conditions to recover Bcd-dependent transcription. We find that a 30 min illumination window spanning n.c. 11 ~ 13 results in severe patterning defects (*Figure 3I*), phenocopying the embryos illuminated during n.c. 13 and 14 (*Figure 3G*). This suggests that Bcd-dependent transcription before n.c. 14 is indispensable for initiating its downstream gene cascade. Shifting the 30 min illumination period to 10 min earlier results in the recovery of proper thoracic segment patterning (*Figure 3J*, arrowhead). As the illumination time window is shifted sequentially earlier, more thorax and mouthparts are successively recovered (*Figure 3K–M*). Last, when the illumination time window is shifted before n.c. 10 (*Figure 3M*), the cuticle patterns become equivalent to those of embryos developing in the dark (*Figure 3C*) or wild-type (*Figure 3B*) and, moreover, the larval hatchability reaches that of dark conditions.

These temporal perturbation experiments reveal that persistent Bcd-dependent transcription activity from n.c.10 is required for robust embryonic patterning, and ultimately embryonic viability. In contrast, formation of the more posterior, thoracic segments is only perturbed when extended illumination covers n.c.13, the critical time window for properly initiating the downstream gene cascade.

## Anterior cell fates require Bcd-dependent transcription for longer duration

Deprivation of Bcd-dependent transcription in late blastoderm stages leads to embryonic lethality, due to errors in cell fate determination. To determine which cell fates are altered by the lack of Bcd function (*Figure 4A*), we fixed illuminated CRY2[high], Bcd[null] embryos at the end of germband extension (GBE, 4 hr after gastrulation initiates) and analyzed the expression of Engrailed (En) and various Hox genes. The most anterior morphological features that can be observed at GBE are the clypeolabrum lobe at the very tip of the embryo and the invagination of stomodeum (*Figure 4B*, red and orange dots, respectively). Blue-light illumination for 10 min before gastrulation impairs the formation of these very anterior structures while the expression of the more posterior En stripes remains

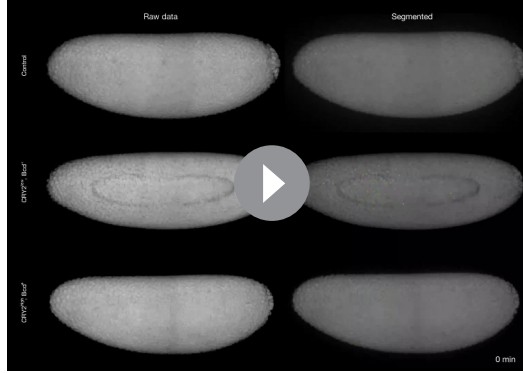

**Video 1.** CRY2::mCh::Bcd expression reduces the transcription activity of hb in the anterior domain in blue light Embryos expressing the hb>MS2, MCP::GFP system were imaged throughout n.c. 13 and 14 with a time resolution of 40 s. Rows 1, 2 and 3 show respectively Control, CRY2[low] and CRY2[high] embryos. The left column represents the raw data whilst the right column shows the segmented MS2 dots (in green) as described in the Materials and methods section.

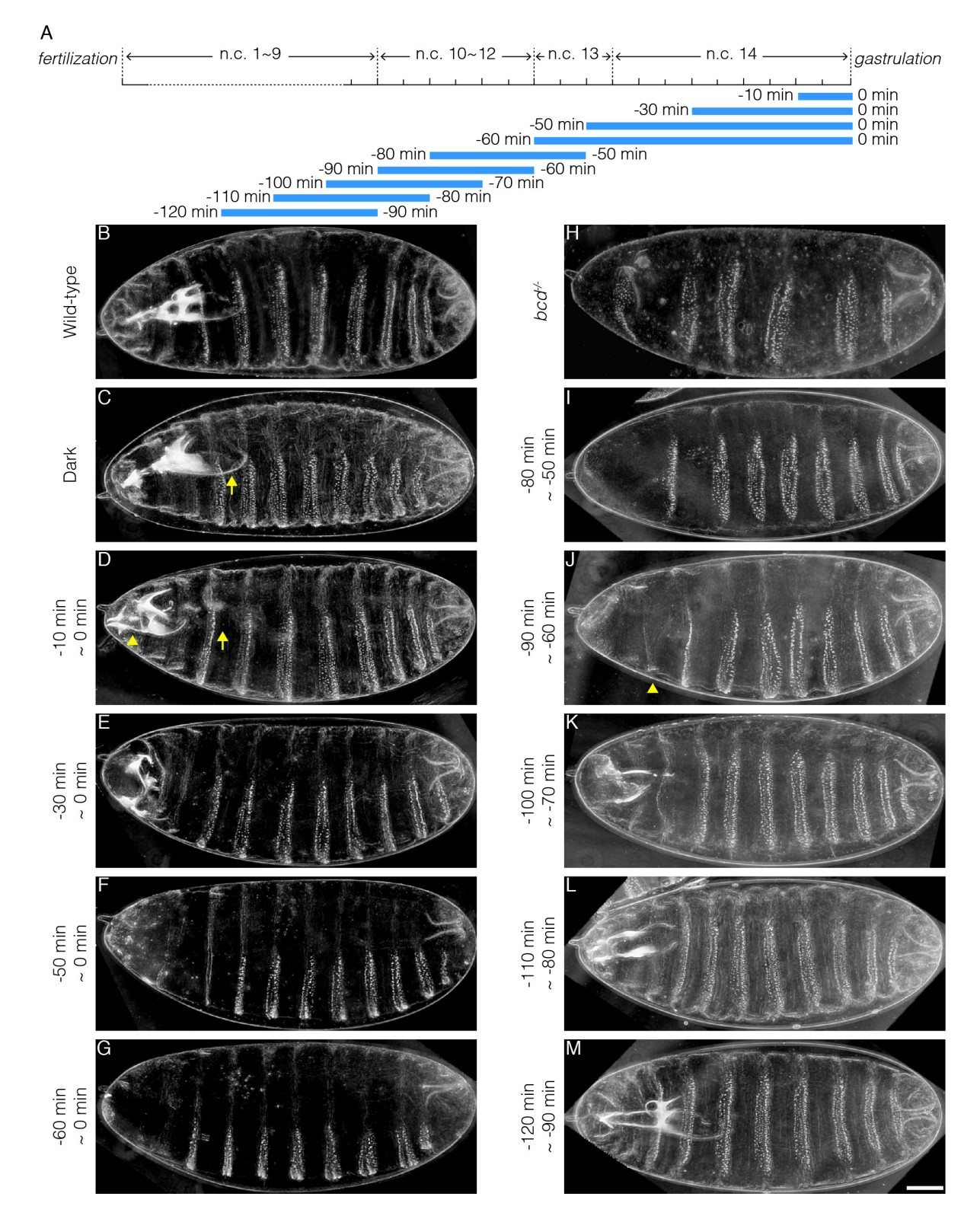

**Figure 3.** Illumination across different time windows causes embryonic lethality with varied severity. (A) Schematic demonstration of illumination time windows. Blue bars indicate illumination while the absence of blue bars indicates dark condition. The onset of gastrulation is defined as time 0. Negative values refer to specific time before gastrulation. The start and the end of illumination is indicated by the number on the left and right side of the blue bars, respectively. (B–K) Cuticle preparation of OreR (B), *bcd*[-/-] (H) and embryos illuminated in different time windows (C–G and I–M). The

*Figure 3 continued on next page*

*Figure 3 continued*
illumination time is indicated on the left of each image. (**C**) Arrow, pharynx wall; (**D**) Arrow, absence of pharynx wall. Arrowhead, missing structures lying between mouth hooks and cephalo-pharyngeal plates. (**J**) Arrowhead, denticle belt of thorax segment. n = 10 embryos per condition. Scale bar, 50 μm.

intact (*Figure 4C*). Extending illumination to 30 min prior to gastrulation, we observe the loss of the next two En stripes marking the mandibular (Mn) and maxillary (Mx) segments (*Figure 4D*). Concomitantly, we observe diminished expression of Deformed, a Hox gene that controls morphogenesis of these same Mn and Mx segments (*Figure 4D'*). This suggests that presumptive Mn and Mx cells commit to alternative cell fates. Similarly, 50 min of illumination prior to gastrulation results in the loss of expression of Sex Comb Reduced (Scr), the hox gene that regulates development of the labial segment (*Figure 4E' and E''*). This correlates with the loss of the En stripe marking this very same segment (*Figure 4E*). Further extending illumination, we observe that anterior cells adopt

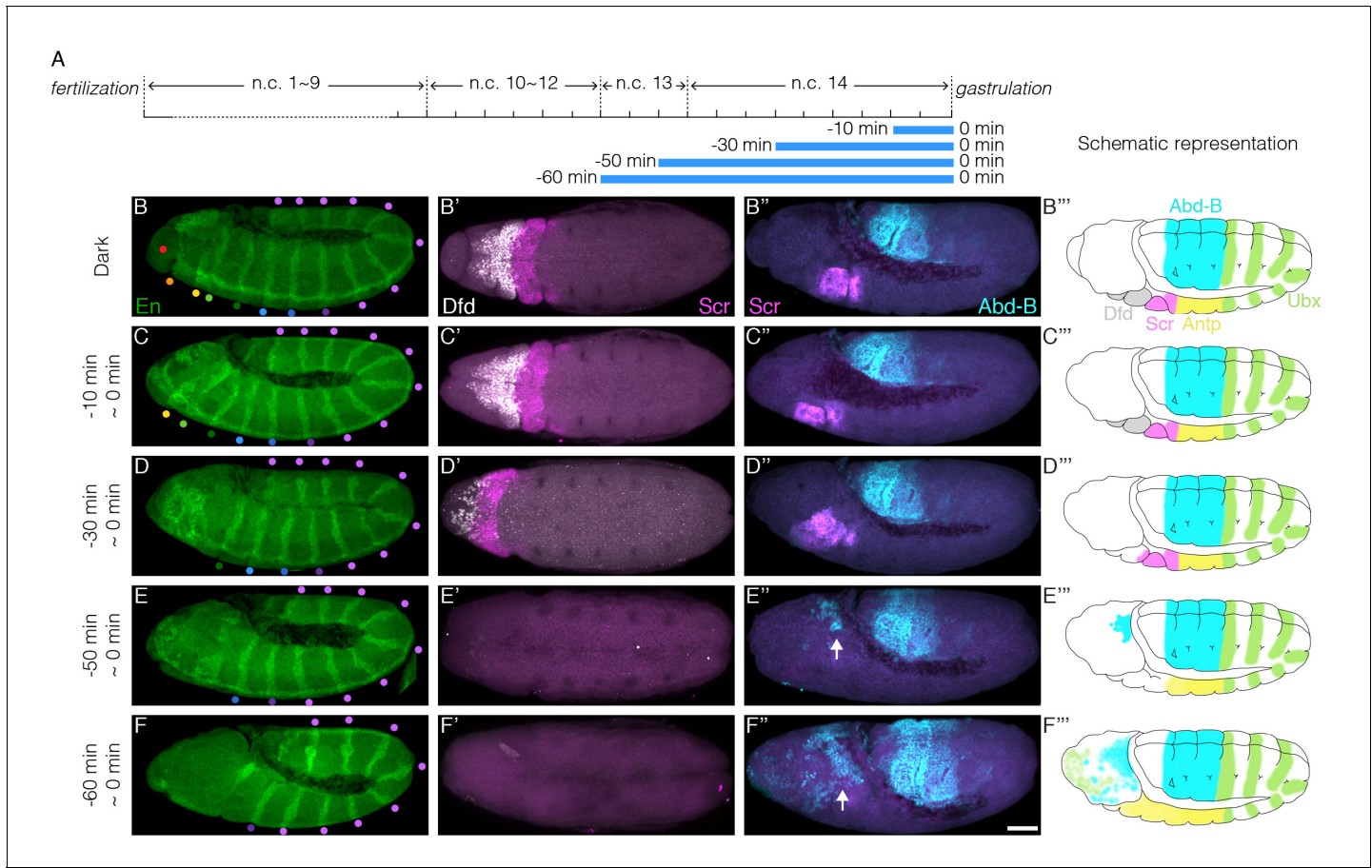

**Figure 4.** Illumination at the end of blastoderm stage causes wrong cell fate determination in anterior embryonic segments. (**A**) Schematic demonstration of illumination time windows at the end of blastoderm stage. Illumination starts at certain time before gastrulation and ends at the onset of gastrulation as indicated by blue bars. (**B–F, B'–F' and B''–F''**) Embryos illuminated in different time windows as indicated at the left side of each panel are fixed by the end of GBE and stained for En (**B–F**), Deformed (Dfd, white) and Scr (magenta) (**B'–F'**), Scr (magenta) and Abd-B (cyan)(**B''–F''**); (**B–F**) Colored dots represent embryonic segments. Red, clypeolabrum; orange, stomodeum; yellow, mandibular lobe; light green, maxillary lobe; dark green, labial lobe; light blue, prothorax; dark blue, mesothorax; dark purple, metathorax; and light purple, abdominal segments. (**E'' and F''**) Arrows, ectopic Abd-B expression. n = 5–10 embryos per condition. Scale bar, 50 μm. (**B'''–F'''**) Schematic representation of hox gene expression maps.

The following figure supplement is available for figure 4:

**Figure supplement 1.** Hox genes expression pattern in increasing illumination time window.

posterior cell fates, reflected in the ectopic expression of Ultrabithorax (Ubx) and Abdomen-B (Abd-B) in anterior regions (*Figure 4E'', F''* and *Figure 4—figure supplement 1F''*, arrows). In the most severe case, we see that all En segments from the anterior to the second thorax (T2) are missing (*Figure 4F*).

Sequential disappearance of En stripes from anterior to posterior correlates with prolonged illumination at the end of the blastoderm stage. This evidence shows that more anterior cell fates require Bcd-dependent transcription to persist until the very end of n.c. 14. In contrast, cells in more posterior regions undergo correct fate decisions at earlier stages, regardless of the Bcd transcription activity at later time points.

## Temporal dissection of Bcd target gene expression

To gain deeper insight into the molecular basis of the spatio-temporal readout dependence on Bcd, we investigated how deprivation of Bcd-dependent transcription at different time windows affects downstream gap gene expression. To this end, we inhibited Bcd-dependent transcription at precise time windows (*Figure 5A*), then fixed these embryos at the end of the blastoderm stage, and stained them for a range of gap genes.

First, we looked at the gap gene hb. Illumination during the last 30 min of the blastoderm stage does not affect Hb in neither protein level nor boundary position (*Figure 5B*), indicating that Hb expression is independent of the newly synthesized mRNA in the last 30 min of n.c. 14. This shows that the Hb protein level remains stable once the *hb* mRNA amount has exceeded a certain threshold value. However, extending illumination throughout n.c. 14 (or even earlier) results in deviations of the Hb anterior pattern (*Figure 5B*, arrows; and *Figure 5—figure supplement 1B4*). It has previously been reported that the Hb pattern due to the activity of the hb stripe enhancer at the posterior border evolves during n.c. 14 from a shallow and broad gradient to a steep and sharp one (*Perry et al., 2011*, *2012*). Here, we show that the mRNA synthesized prior to n.c. 14 is sufficient to support the formation of the broad gradient, as demonstrated by the Hb expression profile in embryos illuminated throughout cycle 14 (*Figure 5B*, ④). Transcriptional activity in the first 15 min of n.c. 14 is hence critical for precise formation of the wild-type Hb expression profile, at both anterior and posterior (driven by the stripe enhancer) borders. Further, continuous illumination throughout n.c. 13 and n.c. 14 severely impairs the anterior Hb expression level (*Figure 5C* and *Figure 5—figure supplement 1B5*), showing that mRNA synthesized during n.c. 13 contributes significantly to the final Hb pattern. Interestingly, recovering the transcriptional activity during the last 30 min of n.c. 14 by reverting the embryos to the dark, results in a partial rescue of Hb anterior expression (*Figure 5C*, arrow), supporting the presence of a *hb* mRNA threshold. Nevertheless, transcription activity in late n.c. 14 fails to recover the stripe enhancer activity (*Figure 5C*, ⑥), potentially due to shifted borders of posterior repressive factors. Next, we tested the contribution of hb transcription prior to n.c. 13. Illumination during n.c. 10–12 results in a steep but narrow Hb gradient, where the posterior boundary shifts anteriorly by ~10% EL (*Figure 5D*, ⑦). Illumination carried out before n.c. 11 results in a recovery of the shape of the Hb gradient when comparing to that of dark conditions, although the posterior border is still anteriorly shifted by 5% EL (*Figure 5D*, ⑧ and *5E–F*, anterior shift indicated by red dashed line). Consistent with this, we observed a significant anterior shift of cephalic furrow position in embryos illuminated during n.c.10–11 compared to dark condition (64.4 ± 1.5% EL in dark and 70.4 ± 1.3% EL in illuminated embryos; *Figure 5—figure supplement 2*). This suggests that the Bcd inputs during n.c. 10–12 are integrated into later decisions on boundary positions. Therefore, we can temporally dissect Bcd-dependent activation of Hb, including separating the contributions from different enhancers.

The expression of the gap genes Kr (58–45% EL) and Kni (45–37% EL) is subjected to fine-tuning by Bcd-dependent transcription (in combination with Hb) (*Hoch et al., 1991*; *Hülskamp et al., 1990*). We find that the anterior boundary of Kr shows very high sensitivity to Bcd transcriptional inhibition - 20 min of illumination during the end of n.c. 14 causes an anterior shift by 3% EL, slightly widening the Kr domain (*Figure 5G*, ②, blue dot). Both the anterior and posterior borders of the Kr expression domain shift anteriorly upon illumination conditions ④, ⑤ and ⑥ (*Figure 5G*), correlating with the shift of the Hb boundary (*Figure 5B*, ④ and *5C*, ⑤ and ⑥). The strongest anterior shift of Kr boundaries is observed when embryos are reverted to dark after early illumination (*Figure 5G*, ⑥ and ⑦), as repressive Hb expression is anteriorly shifted (*Figure 5C*, ⑥ and *5D*, ⑦) while Bcd transcriptional activity is recovered. The Kni anterior boundary shifts in correspondence with the posterior Kr boundary shifts,

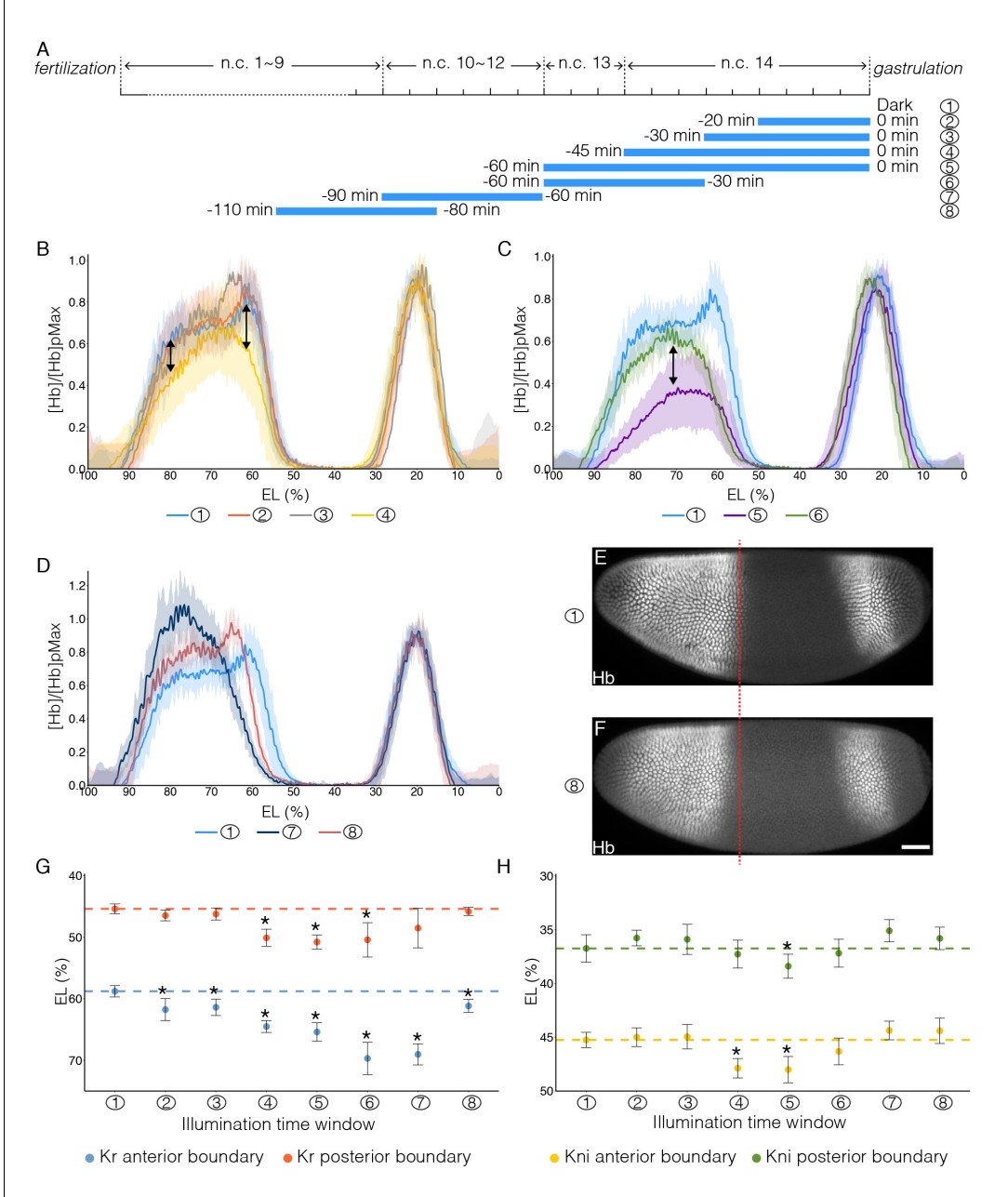

**Figure 5.** Impact of temporally patterned illumination on downstream gene expression. (A) Schematic demonstration of eight different illumination time windows. (B–D) Average Hb intensity normalized to posterior peak values plotted vs. AP position (% EL) in embryos having developed in different temporally patterned illumination. Sample numbers correspond to time windows shown in (A). (B–C) Double-headed arrows point out the changed expression level of Hb. (E and F) Embryos having developed in dark (E) or illuminated for 30 min before n.c.11 (F) are stained for Hb. Red dashed line indicates the position of posterior border of anterior Hb domain of embryo in (E). Scale bar, 50 μm. (G and H) Position of Kr anterior border (G, blue dots), Kr posterior border (G, red dots), Kni anterior border (H, yellow dots) and Kni posterior border (H, green dots) under different temporal illumination as indicated by sample numbers. Error bars indicate s.d.; t-test was used for the statistical evaluation with *p<0.05. A total number of 70, 53 and 59 embryos were analyzed for expression of Hb, Kr and Kni, respectively.

The following figure supplements are available for figure 5:

**Figure supplement 1.** Impact of temporally patterned illumination on downstream gene expression.

**Figure supplement 2.** Cephalic furrow position shifts in different illumination conditions.

*Figure 5 continued*

**Figure supplement 3.** Using optogenetic perturbations to test gap gene models.

while its posterior boundary is much less sensitive (*Figure 5H*). In summary we find that the more anterior expression boundaries are located, the more susceptible they are to temporal inhibition of Bcd-dependent transcription. Further, we find that a tight temporal interplay between Bcd and Hb is required for the precise positioning of gap gene expression boundaries.

## Refining parameter estimation in models of gap gene network

Modelling has been a powerful tool in understanding early *Drosophila* embryo development, with models identifying critical components in the gap gene network required for forming robust gene expression boundaries (*Bieler et al., 2011*; *Jaeger et al., 2004*; *Manu et al., 2009a*, *2009b*; *Perkins et al., 2006*). These models use coupled differential equations to incorporate interactions between gap genes (Hb, Gt, Kr and Kni) and with maternal inputs, along with diffusion and protein degradation. To test how such models perform under temporal perturbation of the Bcd input, we used the model outlined in *Bieler et al. (2011)*, which incorporates the three-dimensional embryo shape, regulatory inputs from Bcd, Cad, Tailless and Huckebein, and the distinct regulatory functions of Hunchback monomers and dimers (Materials and methods and *Figure 5—figure supplement 3A*).

To simulate the temporal Bcd deprivation, the interaction parameters involving Bcd are set to zero at different points within the simulation, corresponding to scenarios ①-⑥ outlined in *Figure 5A*. We first used the optimal parameter set for fitting to gap gene expression profiles from Bieler *et al.*, where the shifted profiles shown in *Figure 5—figure supplement 3B* correspond to scenario ⑤ in *Figure 5A*. The model reproduces the anterior shift in Kni and Kr boundaries upon deactivation of Bcd activity in n.c. 13 and early n.c. 14 (*Figure 5—figure supplement 3C*). However, the behaviour of the anterior domains of Hb and Gt are not consistent with the model predictions (*Figure 5—figure supplement 3B*).

We next used the 21 parameter sets identified in Bieler *et al.* that replicate satisfactorily the wild-type expression profiles for Hb, Gt, Kr and Kni, to explore whether our temporal perturbations can be used to refine the parameter estimations. We assessed the simulation predictions based on the following criteria derived from our optogenetic perturbations: (i) anterior shift in Kr anterior boundary; (ii) anterior shift in anterior boundary of Kni posterior domain; (iii) reduction in Kni anterior expression; (iv) reduction in Gt anterior expression; (v) anterior shift of posterior boundary of Hb anterior domain and reduction in anterior Hb expression. We identified 9 out of 21 parameter sets that qualitatively agreed with at least three of the above criteria. We compared how the average parameter values of this subset differed from the mean parameter values of all 21 parameter sets, (*Figure 5—figure supplement 3D–E*). While the selected subsets did not show much variation in values compared to all 21 parameter sets for interactions affecting the Hb and Kr genes, interactions regulating both Gt and Kni showed larger differences. For example, Bcd plays a more prominent role in activating Gt and Kni autoregulation is reduced in the parameter subset. This suggests that our temporal perturbations can be used to refine models of gap gene segmentation and that gap gene network models need to incorporate Bcd dynamics as well as potential additional interactions within the anterior domain.

## Spatio-temporal atlas of Bcd decoding

We can construct a novel spatio-temporal atlas of Bcd-dependent gene transcription in the early *Drosophila* embryo, with clear time windows for gene activation (*Figure 6A*). A minimal 20 min illumination at the end of n.c. 14 abolishes the expression of the Knirps anterior domain (Kni1) capping the tip of the embryo (*Figure 6B and B'*), as well as the first Giant stripe (Gt1) (*Figure 6C and C'*). Extending the Bcd inactivation across n.c. 14 further inhibits the gap gene expression domains located slightly more posterior, Orthodenticle (Otd) and the second Gt stripe (Gt2) (*Figure 6C'', D and D'*). Recovery of transcription activity in late stages fails to rescue the expression of most of these anterior gap gene domains (*Figure 5—figure supplement 1C6, D6 and F6*). When Bcd-

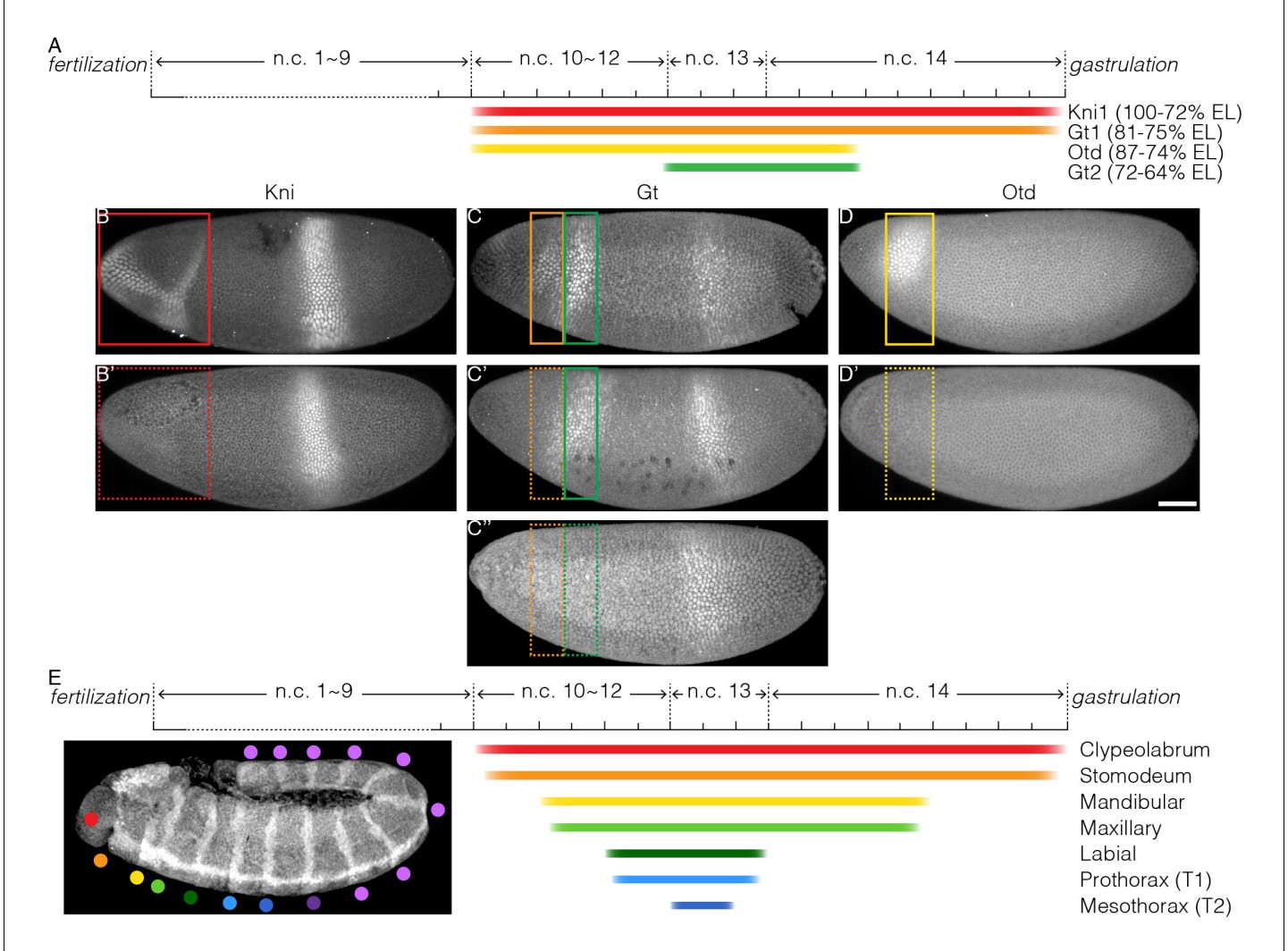

**Figure 6.** Temporal requirement of Bcd-dependent transcription for proper cell fate determination. (**A**) Schematic representation of required time windows of Bcd-dependent transcription for downstream gene expression; color bars indicate the time windows required for Bcd-dependent transcription to be on for correct expression of Kni1 (red), Gt1 (orange), Otd (yellow) and Gt2 (green); (**B–D**) embryos exhibiting the correct gene expression when Bcd-dependent transcription is active in required time window; colored boxes point out the corresponding expression domain; (**B'–D'** and **C''**) embryos showing defects in gene expression when Bcd-dependent transcription is interrupted in required time window; dashed boxes indicate failed expression. Scale bar 50 µm. (**E**) Schematic diagram of temporal interpretation of Bcd morphogen. Colored bars indicate the time windows required for Bcd-dependent transcription for correct cell fate determination in different embryonic segments.

The following figure supplement is available for figure 6:

**Figure supplement 1.** Impeding Bcd-dependent transcription during cycle 14 delays cephalic furrow invagination.

dependent transcription is inhibited during n.c. 10–12 and then embryos returned to dark from n.c. 13, the time window for transcriptional recovery (approximately 1 hr) is insufficient to rescue the expression patterns of Kni1, Gt1 and Otd (*Figure 5—figure supplement 1C7*, D7 and F7). This supports a mechanism whereby Bcd-dependent transcription serves a priming role prior to n.c. 13 to activate anterior segmentation genes at later stages. Such a priming role is stage-specific, as it is not compensated by late recovery of Bcd transcriptional activity. In comparison, illumination for 30 min before n.c. 10 has no apparent impact on segmentation gene expression (*Figure 5—figure supplement 1B8–G8*), corroborating previous results (*Figure 3*) that Bcd-dependent transcription in this time window is dispensable for embryonic viability.

The reliance of the most anterior gap gene expression (Kni1, Gt1, and Otd) on persistent Bcd-dependent transcription may explain our previous observations that inhibiting Bcd transcriptional activity both at very late or early stages impairs the proper formation of the anterior-most segments (*Figure 3D and L*). Any short interruption of Bcd transcriptional activity (from n.c. 10 till the very end of n.c. 14) abolishes the activation or maintenance of these gap gene expression domains. As they lie on top of the hierarchical gene network defining cell fates, their loss alters cell fate decisions and leads to morphological defects. Comparatively, the expression of gap gene domains at posterior positions is less susceptible to perturbations in Bcd function. Consequently, cells arising from these domains still commit to correct fates under Bcd deprivation at early or late stages, as reflected in Hox gene expression patterns. We summarize this spatio-temporally coordinated Bcd interpretation in *Figure 6E*. The more anterior structures, which have cell fates governed by high Bcd concentration, require Bcd for an extended period of time. On the contrary, cells at positions responding to low Bcd require a markedly shorter time window of exposure to Bcd activity to commit to cell fates appropriate for their spatial position.

## Discussion

### Optogenetic manipulation of transcription activity in vivo

Several recent studies have used elegantly designed optogenetic systems to unveil the temporal role of signalling pathways in embryonic cell fate induction (*Johnson et al., 2017*; *Sako et al., 2016*). As a common strategy, a ubiquitously expressed construct is utilized to activate the upstream components of the signalling pathway in a light-responsive manner. In this study, we developed and characterized an optogenetic tool to temporally switch on and off of the transcriptional activity of the morphogen protein Bcd. In contrast to the previous approaches, our light-sensitive protein CRY2::mCh::Bcd is present in the spatial distribution of a native Bcd gradient. While the protein remains transcriptionally active in the dark, blue light-induced conformational change of N-terminal CRY2 abolishes all direct target gene expressions instantaneously. This renders the construct ideal for studying temporal morphogen interpretation. It will be of general interest to further explore if the N-terminal CRY2 exerts transcriptional inhibition when tagged with other transcription factors, so that our optogenetic approach can be more widely applied to understand the temporal readout of spatially distributed transcription factors.

### Mechanism underlying light-induced inhibition of Bcd transcription activity

The following results helped us to narrow down the possible mechanisms underlying light-responsive manipulation of Bcd transcriptional activity. (1) Illumination does not affect the spatial distribution of the protein (*Figure 1J*), suggesting that the normal Bcd diffusion is not hampered. (2) The fact that CRY2::mCh::Bcd inhibits Cad translation in the anterior region of illuminated embryos suggests that CRY2::mCh::Bcd in light can specifically recognize and bind to the Bcd response element in the 3'UTR of Caudal mRNA. This indicates that the homeodomain folding remains intact during illumination. (3) CRY2::mCh::Bcd competes with wild-type Bcd to inhibit target gene transcription in a dominant negative manner, indicating that CRY2::mCh::Bcd in light still binds to its native DNA binding sites. (4) Both cuticle patterns and segmentation gene expression patterns of illuminated embryos show more severe phenotypes compared to Bcd cooperativity mutants (*Lebrecht et al., 2005*), suggesting that the perturbation affects more than just the cooperative binding of Bcd to its target sites. Altogether, these results point to the scenario where light-induced CRY2 conformational change does not alter Bcd specific DNA binding but abolishes its transcription activating potency due to the failure of proper assembly or the release of the transcriptional machinery. The difficulty here lies in specifying the molecular mechanism underlying such an inhibitory effect. Roles of the Bcd N-terminal domain in mediating protein interactions were elucidated previously (*Fu et al., 2003*; *Ma et al., 1996*). Here, it is unclear whether it is Bcd-Bcd interaction, Bcd interaction with its co-activators and/or other general transcription cofactors that is inhibited. Hopefully, future structural studies on Cryptochromes light-responsive conformational changes will help us to clarify this issue.

## The temporal pattern of Bcd signal integration is spatially inhomogeneous

The continuous transitions between cell states in a developing embryo are governed by both the transcriptional landscape and the transcriptional history of the cells. Therefore, not only the local concentrations of the combinatorial transcription factors but also the timing and duration of their presence are essential for the correct cell fate decisions. Bcd in the early fruit fly embryo provides the very initial positional cues along the antero-posterior axis, differentiating cells spanning ~10 embryonic segments (*Driever et al., 1989*). The gradient profile of Bcd is highly dynamic throughout the blastoderm stage, with no clear steady-state (*Little et al., 2011*). Such dynamics make the temporal aspects of Bcd interpretation critical for precise decoding of the morphogen. A previous study quantified the temporal evolution of Bcd dosage at several gap gene expression boundaries, and deduced that the absolute Bcd concentration is read out during early n.c. 14 (*Liu et al., 2013*). In comparison, data-driven models propose a much wider time window for Bcd readout, as decoding the gradient earlier in development (*Bergmann et al., 2007*) or during its degradation in n.c. 14 (*Verd et al., 2017*) gives rise to more accurate predictions of gap gene expression patterns. Direct testing of these ideas has previously been difficult due to a lack of temporal manipulation of Bcd activity.

Consistent with previous studies, we find that the time window ranging from n.c. 13 to early n.c. 14 is indeed most critical for Bcd-dependent patterning of downstream genes; deprivation of Bcd transcriptional activity during this period leads to severe patterning defects in all the Bcd-dependent embryonic regions, from the most anterior to mesothoracic segment (*Figure 3G and I*). On the other hand, Bcd activity in this critical time window is sufficient for the proper cell fate induction in mesothorax, regardless of Bcd activity in rest of the time points throughout the blastoderm stage. Further, the more anterior the cells locate, the wider time window of Bcd activity is required (*Figure 3*), as in the extreme case, the proper formation of the most 'needy' clypeolabrum segment, patterned by the highest Bcd concentration, requires Bcd-dependent transcription from n.c. 10 to the very end of n.c. 14 (*Figure 4B–C*).

Similar correlations between morphogen concentration and required duration have been observed in Shh patterning of the vertebrate neural tube (*Dessaud et al., 2007*). Shh signalling is temporally integrated by a genetic feedback loop that leads to desensitization of cells to Shh signal over time. In this way, the duration of Shh input is translated into differential gene expression. Considering the rapid establishment Bcd target gene expression patterns, it is unlikely that the same mechanism is utilized here. Our quantitative analysis on gap gene expression stand in line with two alternative mechanisms underlying such temporal integration: (1) The anterior gap genes have slower transcription rates than the posterior gap genes, therefore they require Bcd for longer duration. A recent study has proposed a role of transcription kinetics in shaping the timing as well as the spatial range of morphogen response (*Dubrulle et al., 2015*). Here, we find that the mRNA production and protein turnover of the anterior gap genes are in a tight balance. Once this balance is tipped by inhibiting transcription even for a short period of time, the protein expression can no longer be maintained (*Figure 6A–D*). In contrast, inhibiting the transcription of more posterior genes, such as hb, in mid to late n.c. 14 does not affect their expression, potentially due to an excessive mRNA pool. (2) Bcd-dependent transcription serves as a priming role in early stages for proper expression of anterior gap genes. Restoring native Bcd dosage in later stages fails to initiate *bona fide* gene expression in the very anterior domains, suggesting that cells are no longer competent to adopt anterior-most cell fates. Whether Bcd primes cell competency by chromosomal remodeling to increase accessibility (*Blythe and Wieschaus, 2016*) or inhibiting the otherwise ectopically expressed repressive factors must be subjected to further investigation. These recent results, including those outlined in this paper, suggest that models of gap gene expression need to incorporate temporal variation more carefully. In particular, anterior gap genes depend on Bcd for longer than more posterior genes, which is not reflected in current models. Our optogenetic tool allows precise and subtle perturbations to be performed, allowing a more systematic test of the models for gap gene expression.

## Phenotypic resemblance between hypomorphic bcd alleles and short temporal perturbations

Three anterior embryonic regions with different sensitivity for bcd reduction can be distinguished, that is, the preantennal head region, the antennal and gnathocephalic head segments, and the

thorax. While the weakest *bcd* alleles only affect the first region, defects sequentially extend to the second and third regions with increased allelic strength (*Frohnhöfer and Nüsslein-Volhard, 1986*). Interestingly, the same distinction can be made by temporally perturbing Bcd activity, with prolonged transcriptional inhibition giving rise to defects in sequentially extending regions.

The molecular properties of these hypomorphic *bcd* alleles affect either the DNA binding affinity (e.g. $bcd^{E3}$, point mutation in homeodomain) or the activation potency (e.g. $bcd^{2-13}$, partial deletion of the C-terminal activation domain) of the Bcd protein (*Struhl et al., 1989*). The resemblance between hypomorphic *bcd* alleles and short temporal perturbations demonstrates that constitutively low *bcd* activity is functionally equivalent to normal bcd activity with reduced active duration; both are sufficient to guide proper cell differentiation trajectories in the mid region of the embryo where target gene binding affinities are high. In comparison, in the more anterior region where target genes have low affinities, both wild-type level bcd activity and long duration are necessary for anterior induction. It is noteworthy that duplicated posterior structures only appear in strong allelic mutants but not in embryos subjected to long illumination due to a normal Cad gradient in our experiments. It would be interesting to understand whether hypomorphic *bcd* alleles can suppress Cad translation.

## Temporal interpretation ensures developmental precision by buffering noises

What role does temporal coordination of morphogen interpretation play in achieving precise cell fate determination? In *Drosophila* - as in all long germband insects - the embryonic segments emerge simultaneously (*Sander, 1976*). Bcd-dependent cell fate specification appears to move from posterior towards the anterior in a sequential manner. The most posterior cells 'lock-in' to their correct fate decisions by early cycle 14, becoming refractory to further alterations of Bcd dosage. Meanwhile, more anterior genes remain sensitive to Bcd dosage for much longer periods. Therefore, this mode of temporal interpretation can cope better with a temporally dynamic morphogen gradient or fluctuations in local concentration. This temporal sequence of boundary determination may help to assure robustness of gene boundary specification within such a short developmental time.

## Tight coupling between transcription and morphogenesis

Our optogenetic transcriptional manipulation approach brings further insights into how transcription is tightly coupled to cell specification and could be extrapolated to other analyses as morphological movements during embryogenesis. For example, we find that Bcd-dependent transcription not only determines the position of cephalic furrow invagination (*Vincent et al., 1997*) but also dictates the timing of when this

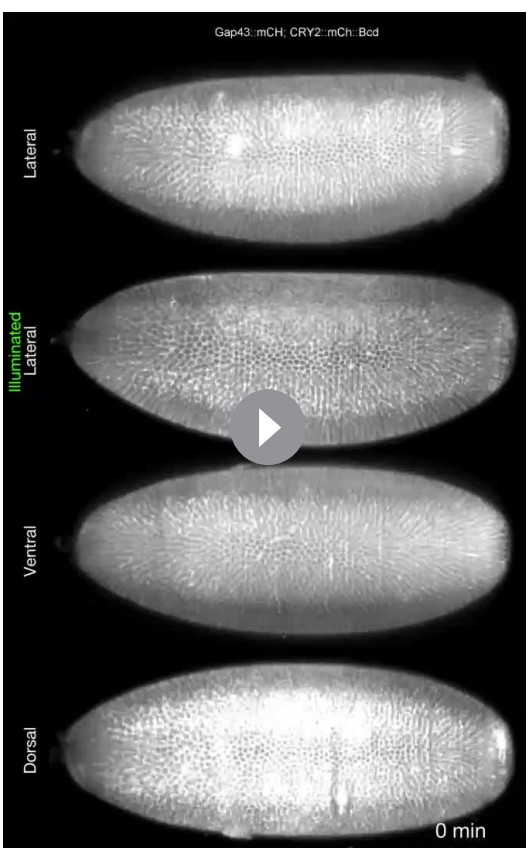

**Video 2.** Bilateral cephalic furrow formation is uncoupled by single sided illumination Embryo expressing CRY2::mCh::Bcd and Gap43::mCh mounted on a custom-built Light-Sheet microscope. The embryo was mounted during n.c. 13 and Gap43::mCh was used to follow its developmental stage. Five minutes after the start of cellularization, one lateral side of the embryo was illuminated for 20 min, starting from its most apical section to 70 μm deep (0.7 μm interval). Whole embryo recording (561 nm laser) and hemi embryo illumination (488 nm laser) were done simultaneously with a time resolution of 30 s. The top two panels show the lateral sides of the embryo, the second panel being the illuminated one. The bottom two panels show the ventral and the dorsal, respectively. n = 7 embryos per condition.

tissue remodeling occurs (*Figure 6—figure supplement 1*). Finally, coupling optogenetic manipulations with light-sheet microscopy, we can spatially restrain transcriptional inhibition to one lateral side of the embryo. This locally alters cell fates, resulting in morphological defects – such as in cephalic furrow formation - while the contralateral side remains intact (*Video 2*). Spatio-temporal manipulation of Bcd activity in the early embryo is thus an exciting avenue for future study.

## Materials and methods

### Fly stocks and genetics

A CRY2::mCh::Bcd fusion construct was generated by replacing the eGFP sequence in the P[*egfp-bcd*] vector (*Gregor et al., 2007a*) by the CRY2::mCherry sequence. The original vector was digested with NheI and SphI to remove eGFP and the CRY2::mCherry sequence was amplified by PCR from the AddGene plasmid #26866 (*Kennedy et al., 2010*) using primers appended with NheI and SphI restriction sites. The resulting **P[*cry2-mcherry-bcd*]** construct contained *bcd* natural 5' and 3' UTRs as well as upstream enhancers. P-element transformation was carried out by BestGene Inc. and seven transgenic lines on the 3$^{rd}$ chromosome were recovered. Two different lines expressing CRY2::mCh::Bcd respectively at low (CRY2$^{low}$) and high (CRY2$^{high}$) levels (mCherry fluorescent intensity) were employed throughout our analyses. Transcription and mRNA localization in the transformed flies was expected to recapitulate those of endogenous *bcd*.

A *bcd* knockout fly line was generated by CRISPR-mediated insertion of a MiMIC cassette (*Venken et al., 2011*) (injection performed by GenetiVision). The progenies from homozygous females phenocopied *bcd$^{E1}$* developmental defects (*Frohnhöfer and Nüsslein-Volhard, 1986*). The CRY2::mCh::Bcd (CRY2$^{high}$) transgenic was recombined with the *bcd* knockout allele to establish a stable line. All experiments were carried out with this fly line unless otherwise stated.

MCP::GFP and *hb*>MS2-yellow fly lines (*Garcia et al., 2013*) used in the quantitative live imaging of transcription were kindly provided by Thomas Gregor (Princeton University). Virgin females expressing both MCP::GFP and CRY2::mCh::Bcd were crossed with males of the reporter line *hb*>MS2-yellow to collect embryos for imaging.

A pCaSpeR4 vector containing the construct pNOS-NoNLS-MCP-GFP-αTub3'UTR was kindly provided by Thomas Gregor (Princeton University). The GFP sequence is excised and replaced by mCherry using the Gibson Assembly strategy (NEB). The construct pCaSpeR4-pNOS-NoNLS-MCP-mCh-αTub3'UTR was randomly inserted into the fly genome using P-element transformation (Best-Gene). A line with II chromosome insertion was crossed to CRY2-mCh-Bcd on the III chromosome to establish a stable line *nos*>MCP::mCh; CRY2::mCh::Bcd. Virgin females of this line were crossed to males of the reporter line hb>MS2-yellow to produce embryos for quantitative live imaging.

Additionally, the fly line nanos>Gap43::mCh (*Martin et al., 2010*) was used for light-sheet imaging.

### Temporally patterned illumination

Duration of syncytial nuclear cycles were evaluated by imaging embryos on a bright-field stereomicroscope at 25°C. The blastoderm stage (from fertilization to the onset of gastrulation) spans 2.5 hr. The last syncytial cycle (n.c. 14), demarcated by the last division wave and the first sign of gastrulation, lasts about 45 min. The penultimate cycle (n.c. 13) lasts 15 min, while the duration of the previous three cycles (n.c. 10–12) is about 30 min in total. To maintain dark condition, embryos were observed or imaged with all light sources covered by amber paper (*i.e.* blocking blue light). To induce the conformational change of the CRY2 protein, we illuminated the embryos on a Nikon LED light base at 488 nm wavelength. The light intensity was measured with an intensity power meter. All experiments were carried out at 4.0 mW unless otherwise stated. For temporally patterned illumination, blastoderm stage embryos were selected in the dark condition and exposed to light. The illumination treatment of each embryo was recorded. The illumination duration was timed from the moment of light exposure until the onset of gastrulation. When the illumination treatment was followed by dark recovery, we continued the timing by observing embryos with a light source covered with amber paper until gastrulation initiated.

## Immunostaining

Embryos at the desired stages were dechorionated by household bleach and fixed in heptane saturated by 37% paraformaldehyde (PFA) for 1 hr. The vitelline membrane was subsequently manually removed. Prior to incubation with primary antibodies, embryos were blocked with 10% BSA in PBS. Image-iTFX signal enhancer was used as blocking reagent instead of 10% BSA for Cad staining. Antibodies used were rabbit anti-mCherry (1:100, Abcam Cat# ab183628 RRID:AB_2650480), rat anti-Caudal (1:100), guinea pig anti-Hb (1:2000), rabbit anti-Gt (1:800), guinea pig anti-Kr (1:800), guinea pig anti-Kni (1:800), guinea pig anti-Eve (1:800), guinea pig anti-Otd (1:800), mouse anti En (1:100, DSHB Cat# 4D9 anti-engrailed/invected RRID:AB_528224), rabbit anti-Dfd (1:100), mouse anti-Scr (1:10, DSHB Cat# anti-Scr 6H4.1 RRID:AB_528462), mouse anti-Antp (1:10, DSHB Cat# anti-Antp 8C11 RRID:AB_528083), mouse anti-Ubx (1:100, DSHB Cat# UBX/ABD-A FP6.87 RRID:AB_10660834) and mouse anti-Abd-B (1:100, DSHB Cat# anti-ABD-B (1A2E9) RRID:AB_528061).Primary antibodies were detected with Alexa Fluor-labelled secondary antibodies (1:500; LifeTech). Embryos were co-stained with Phalloidin conjugated with Alexa Fluor for staging purpose. Embryos were mounted in AquaMount (PolySciences, Inc.) and imaged on a Zeiss LSM710 microscope with a C-Apochromat 40x/1.2 NA water-immersion objective. Cad antibody was kindly provided by Eric Wieschaus. Hb, Gt, Kr, Kni and Eve antibodies were gifts from Johannes Jaeger. Otd was kindly given by Tiffany Cook. Last, Dfd antibody was a gift from Thomas C. Kaufman.

## Gradient quantification

Images were projected using a maximum intensity projection and then nuclei segmented using Ilastik (*Sommer et al., 2011*). Nuclei were binned into 5 mm spatial steps along the anterior-posterior axis using Matlab (RRID:SCR_001622). Bcd nuclear intensity plots were created after background subtraction using morphological opening. Profiles fitted as described in Liu (et al., 2013).

## Live imaging of MS2 RNA reporter

Embryos were collected, dechorionated and mounted on a MatTek dish under dark condition. For the imaging of MCP::GFP lines, the embryos were imaged on a custom-built Spinning Disc microscope with a Nikon Apo 40X/1.25 water-immersion objective. The pixel size is 409 nm and the image resolution is 1024 × 1024 pixels. At each time point a stack of 35 images separated by 3 µm was acquired. The temporal resolution was 40 s. For the imaging of MCP::mCh lines, the embryos were imaged on a Zeiss LSM710 microscope with a C-Apochromat 40x/1.2 NA water-immersion objective. A laser wavelength of 561 nm was used to excite MCP::mCh and an LED light source was used to illuminate the embryo to induce conformational change of CRY2::mCh::Bcd. Pixel size is 346 nm and a rectangular section with image resolution of 750 × 220 pixels was acquired in the middle region of the embryos (*Figure 2—figure supplement 3A*). At each time point a stack of 20 images separated by 1 µm was acquired and time resolution is 30 s. Each Z-stack for each time point was Z-projected (maximum intensity) prior to subsequent quantification.

## Quantification of MS2 RNA reporter

Nascent hb transcripts were detected from live-imaging data using both ImageJ (*Schindelin et al., 2012*) and Matlab. Each Z-stack for each time point was Z-projected (maximum intensity). In order to automate the tracking and subsequent analysis, the following parameters were manually measured from the projections: lowest intensity value, spot size (matrix size, $N \times N$), maximum displacement distance within two frames and intensity difference between MS2 dots and surrounding background. The embryo boundary was automatically traced and extracted from projected images by using the 'bwboundaries' function (*Gonzalez et al., 2004*). This measurement was later used to exclude false positive and to calculate the coordinates for each MS2 dots. At each time frame, MS2 dots were localized by applying a threshold corresponding to the lowest intensity value. Pixels showing an intensity higher than this threshold were selected. Following the selection, the pixels out of the embryo boundary were excluded. Next, a regional intensity comparison was performed by comparing the intensity of the selected pixels to its surrounding pixels within a $(N+2) \times (N+2)$ matrix window. The intensity difference between the selected pixel and the average intensity of its neighbours was calculated and pixels having a value higher than the approximate intensity difference between MS2 dots and surrounding background were selected. Central coordinates, averaged

intensity value and AP position were recorded for subsequent quantification. Tracking of MS2 dots used the maximum MS2 dots displacement distance calculated above to define a displacement range. Hence, a MS2 dot at the frame t was connected to the nearest dot at the frame t + 1 within the displacement range. Last, we excluded dot tracks lasting less than 160 s as it is shorter than a complete hb transcription event. The probability distribution of MS2 dots persistence in the anterior and posterior domains was calculated. The anterior domain was subdivided into five regions (100–75, 75–70, 70–65, 65–60 and 60–40 %EL) to refine the analysis.

## Chromatin immunoprecipitation and qPCR

Late n.c. 14 embryos aged in the dark or exposed to light (at 4 mW) for 45 min were dechorionated in household bleach. Embryos were crosslinked for 15 min in a solution containing 2 ml of PBS, 6 ml of Heptane and 180 µl of 20% paraformaldehyde. Embryos were transferred to a 1.5 ml tube and the crosslinking was quenched with the addition of 125 mM glycine in PBS 15 min after the start of fixation. ChIP samples were essentially prepared as described in *Blythe and Wieschaus (2015)*. Sonication was performed on a Sartorius stedim Labsonic M with a microtip horn. An input control corresponding to 2% of the volume per reaction was taken after sonication. Immunoprecipatations (IPs) were performed with a mCherry antibody (Clonetech #632496) for 15 hr at 4°C. ChIPped DNA were extracted with a Qiaquick spin column (Qiagen). Real-time quantitative PCR was performed using SYBR Green Assay (Thermo Fisher Scientific) on a Bio-Rad CFX96 Real-time system. Four primer pairs were used, respectively probing Hb (P2 enhancer), Gt, Otd and Kr. Additionally, a primer pair was designed 1 kb downstream of the P2 enhancer (HbP2 +1 kb) to be used as a negative control. The percent input method ($100 * 2^{(\text{Adjusted input} - Ct(IP))}$) was used to normalize ChIP-qPCR data. Further, the relative % input for illuminated embryos was calculated as compared to the embryos in dark and summarized in a bar chart. In the case of the HbP2 +1 kb, data were compared to HbP2 results. Three independent replicates have been performed for each primer sets and the significance was calculated with a standard t-test.

## Cuticle preparation

Embryos subjected to temporally patterned illumination were allowed to develop until the end of embryogenesis. The embryos were then dechorionated and incubated into a mixture of Hoyer's medium and Lactic acid in a 1:1 ratio at 65°C between an imaging slide and a cover slip. For an exhaustive description of the method used see *Alexandre (2008)*.

## Gap gene profile quantification

Confocal Z-stack images were Z-projected (maximum intensity) in Fiji for further analysis. Images were rotated to orient embryos anterior left and dorsal up and rescaled to same embryo length and width. Intensity profiles along the antero-posterior axis were measured in Fiji (RRID:SCR_002285). For Hb, the intensity profile was normalized to the peak value of the posterior domain. To determine the boundary positions of Kr and Kni expression domains, we plotted the intensity profiles along the AP axis and defined the boundary at the position with intensity equals to half of the peak value.

## Mathematical modelling

The model analyzed is from *Bieler et al. (2011)*, with code available at http://3d-flies.epfl.ch/. The initial conditions for the simulations are set at the beginning of cycle 12 and the dynamics are modelled over the next 90 min. A matrix, $T_{ij}$, defines the interaction parameters both between the gap genes (Hb, Gt, Kr, and Kni) and between the gap genes and morphogenetic inputs (Bcd, Cad, Huckebein (Hkb) and Tailless (Tll)). At times corresponding to scenarios ①-⑥ in *Figure 5A*, the interaction parameters $T_{Bcd,j}$ are set to zero. The shifts in $T_{Bcd,j}$ are assumed to be instantaneous. Scoring to the criteria outlined in the text is done manually. The parameter value variation between a subset, $p_i^{subset}$, and the average of all n = 21 parameter sets, $p_i^{all}$, was scored for each parameter using $z_i = \frac{\langle p_i^{all}\rangle - \langle p_i^{subset}\rangle}{\sigma_i^{all}/\sqrt{n}}$, *Figure 5—figure supplement 3E*. The z-score shown in *Figure 5—figure supplement 3E* corresponds to all nine datasets that agree with at least three of the criteria for matching our optogenetic perturbations.

## Hemi embryo illumination on a light-sheet microscope

Embryos were dechorionated as described earlier and mounted in 1% agarose on a Fluorinated ethylenepropylene (FEP) capillary (TEF-CAP, #AWG18LW-FEP). Embryos were then imaged on a custom-built Light-Sheet Microscope with a Nikon Apo 1 WD 25X/1.10 water-dipping objective. Embryos expressing CRY2::mCh::Bcd and the membrane marker Gap43::mCh were recorded using a 561 nm excitation laser. The pixel size was 510 nm and the image resolution was 1024 × 1024 pixels. At each time point a stack of 100 images separated by 2.5 µm was acquired. The temporal resolution was 30 s. This illumination setup mimics dark condition as no morphological defects were observed. To stimulate CRY2 conformational change in just one half of the embryo, a region on one lateral side of the embryo was defined from the most apical section up to 70 µm down. A stack of 100 sections separated by 0.7 µm were illuminated. The illumination was carried out from the beginning of the n.c. 14 (5 min after the establishment of the cellularization furrow) for 20 min with a 488 nm laser. Following illumination, embryos were imaged solely with the 561 nm excitation laser.

## Acknowledgements

We thank Eric Wieschaus, Thomas Gregor, Johannes Jaeger, Tiffany Cook, Thomas Kaufman and James Sharpe for sharing precious reagents. We acknowledge Eric Wieschaus, Enrique Martin-Blanco, James Briscoe, Philip Ingham, Shelby Blythe and all Saunders' lab members for fruitful discussions. This work was supported by the National Research Foundation Singapore under an NRF Fellowship to TES (NRF2012NRF-NRFF001-094).

## Additional information

### Funding

| Funder | Author |
|---|---|
| National Research Foundation Singapore | Timothy E Saunders |

The funders had no role in study design, data collection and interpretation, or the decision to submit the work for publication.

### Author contributions

AH, CA, Conceptualization, Data curation, Formal analysis, Investigation, Visualization, Methodology, Writing—original draft, Writing—review and editing; SZ, Formal analysis, Visualization; NST, Resources; TES, Conceptualization, Supervision, Funding acquisition, Writing—original draft, Writing—review and editing

### Author ORCIDs

Anqi Huang, http://orcid.org/0000-0003-0551-1160
Christopher Amourda, http://orcid.org/0000-0001-5173-0159
Timothy E Saunders, http://orcid.org/0000-0001-5755-0060

### Ethics

Animal experimentation: This study was performed in strict accordance with the recommendations of the Agri-Food and Veterinary Authority of Singapore (Ref. NEA/CLU/16-0008).

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
