## [Decision Letter]

Thank you for submitting your article "Decoding temporal interpretation of the morphogen Bicoid in the early *Drosophila* embryo" for consideration by *eLife*. Your article has been reviewed by three peer reviewers, and the evaluation has been overseen by a Reviewing Editor and Didier Stainier as the Senior Editor. The following individual involved in review of your submission has agreed to reveal his identity: Jun Ma (Reviewer #3).

The reviewers have discussed the reviews with one another and the Reviewing Editor has drafted this decision to help you prepare a revised submission.

As you can see below, all three reviewers recommended your study and were particularly excited about your ability to manipulate temporal parameters of the Bicoid gradient.

The most significant recommendation was to perform some theoretical/computational analysis of the gap gene network, to examine your results, or call for the revision of existing models.

*Reviewer #1:*

For the last 30 years, the Bicoid gradient has been the most studied example of a morphogenetic gradient. It would thus seem difficult to find new important details that have been missed by these studies that went deep into molecular mechanisms and models of gene regulation. Yet, this paper appears to provide just that: not a huge amount of conceptual novelty, but a powerful way to manipulate temporal parameters of the gradient using a very clever method to render (some of) the activity of the protein light dependent.

The authors achieved this by fusing the Bicoid protein in the context of its own rescue construct to a light sensitive moiety. Interestingly, this appears to only affect the transcriptional activity of the protein and not its DNA binding or its RNA regulatory capacity: the activation domain might be more sensitive, which is surprising but the data show a full rescue without light and no rescue with light. This allowed the authors to stop activation by Bicoid at various times during development. They conclude that the temporal pattern of Bcd activation is important, which appears to be in slight contradiction with the work of Thomas Gregor who also quantified very carefully the effect of Bicoid but could not manipulate it so precisely. It appears that genes that require higher Bcd concentration require Bcd for longer duration, while exposed to low Bcd (whose targets have high affinity Bicoid binding sites) can be specified with short early exposure.

Other than that, it's the most novel contribution to the study of *Drosophila* AP patterning that I've seen in years. I'm not sure they solve a lot of the issues they raise, but the experiments and data are pretty interesting. I think understanding temporal contributions of transcription to cell fate are interesting, and it is great that the authors used this system to achieve this and their results are convincing: this is a powerful tool and they might be able to obtain new concepts with it.

*Reviewer #2:*

Huang et al. use an optogenetic system to investigate the temporal requirement for Bcd activity in *Drosophila* blastoderm patterning. The authors find that Bcd is required from 1h post-fertilisation (cell cycle 10) and that specification of anterior regions requires longer durations of Bcd activity than posterior regions. The authors used the approach to generate a map of the temporal requirements for Bcd activity.

An optogenetically controllable Bcd is an elegant and powerful reagent which provides a new perspective to this well established system. As the textbook example of a morphogen, this study will be of interest not only to the Bicoid field but to the wider developmental biology community. Overall the experiments are convincing and well documented and the manuscript is clear and accessible. Defining differential time requirements for different target genes and demonstrating a role of Bcd prior to cell cycle 13 are important results that will lead to a re-evaluation of currently favored models of Bcd dependent patterning.

The authors characterize the CRY2::mCh::Bcd in Figure 1 and 2. Additional information would be helpful and some insight into the mechanism would be informative. This seems particularly important, given the dominant negative activity of the construct.

The authors suggest that illuminated embryos "resemble" the bcd knockout phenotype (subsection “Precise temporal control of Bcd-dependent transcription via optogenetic manipulation”, second paragraph). In Figure 1—figure – supplement 2 the authors show that the phenotypes are not identical. This should be highlighted earlier in the text and the phenotypic comparison given more interpretation, to help readers not expert in the system.

The mechanism by which the cryptochrome cassette blocks Bcd activity is unclear. The authors note that spatial distribution of Bcd and DNA binding activity is unaffected and the Caudal gradient is also unaltered. These experiments would be strengthened by controls in which Bcd and Cad gradients are disrupted, to gauge the sensitivity of the assays. The genotypes of these embryos should also be clearly stated in the figure legend.

Experiments that directly test how quickly Bcd dependent transcription is lost in illuminated conditions and recovered following cessation of illumination would strengthen the interpretation of Figure 3.

Figure 4 demonstrates the sequential loss of identities from the anterior as the illumination times are increased. It would be helpful to have a schematic of gene expression in this figure. While the loss of anterior gene expression is evident, the anterior expansion of more posterior fates (as would be expected in a homeotic-like transformation) is less evident.

The data in Figure 5 and Figure 6 provide a detailed quantitative description of the temporal requirements for Bcd activity. These are novel and interesting findings that will help move the Bcd patterning field forward. Given the authors' expertise and the availability of dynamical models of the Gap gene system (from Reinitz, Jaeger and others) it seems a missed opportunity not to test whether in silico simulations using the existing models (interrupting Bcd activity during defined time windows) recapitulates the experimental results. Although developing an entirely new mathematical model might be beyond the scope of the current study, testing whether the data fit existing models would indicate whether new models (or new parameters for existing models) will be necessary. Related to this, the authors might want to extend their Discussion to consider the involvement of the dynamics of the Bcd controlled transcriptional network (as well as the kinetics of individual gene responses) in the differential sensitivity to durations of Bcd activity. This would allow a more direct comparison with the vertebrate neural tube where the dynamics of the Shh responsive transcriptional network are implicated.

*Reviewer #3:*

Huang et al. describe an optogenetic manipulation approach for investigating the temporal requirement of the *Drosophila* morphogen Bicoid (Bcd) for embryonic development. Their general conclusion is that cell fates dependent on higher Bcd concentration require the Bcd input for a longer duration of time. The concepts are interesting and most of the experiments are well controlled. The findings reported here are useful to the field. There are a few aspects that can be improved.

While the observed effects of light exposure on embryonic development and gene expression are convincing, the underlying mechanisms remain relatively unclear. This is because precisely how the CRY2::mCh::Bcd fusion protein becomes inactivated at the molecular or biochemical level has not been established. The authors describe experiments suggesting that neither the ability of Bcd in translation inhibition (monitored by Cad protein gradient profile) nor its DNA binding (monitored by ChIP) was affected. It is possible that, as suggested by the authors, the N-terminal fusion of CYR2 to Bcd may cause a light-induced conformational change that inhibits the transcriptional activity of Bcd (since its N-terminus does contain a regulatory motif modulating its transcription activity). But I am not fully convinced that the ChIP approach can sensitively and unambiguously rule out an effect on DNA binding. The fact that the expression domains of hb and Kr exhibit an anterior shift (Figure 5) is consistent with a potential effect on Bcd DNA binding (or Bcd gradient profile might also be altered – Figure 1). While the distinction between DNA binding and transcriptional activation is crucial to understanding Bcd action as a transcription factor and several reports have recently described perturbations specifically altering the activation potency of Bcd, I believe the current manuscript can be written in a way that sidesteps this issue; but if the authors prefer to discuss it, they may need to provide a more detailed Discussion. Interestingly altering the activating potency of Bcd is known to cause changes in the shutdown timing of hb transcription and the hb expression level at nc14 in a specific manner. Whether and, if so, how these results are related to the current findings (particularly with regard to the persistence of transcription at nc13) should be discussed to give the reader a more comprehensive update of the current knowledge.

It is generally acknowledged that weaker bcd alleles tend to affect preferentially the formation of the most anterior structures, suggesting that such structures are more sensitive to perturbations of the Bcd input. It would be useful to discuss how this study may (or may not) provide mechanistic insights into those earlier findings.

It is useful to provide a brief discussion on the estimated time scales of both illumination-induced inactivation and "reactivation" in experimental setting described in the current study.

---

## [Author Response]

*The most significant recommendation was to perform some theoretical/computational analysis of the gap gene network, to examine your results, or call for the revision of existing models.*

We would like to thank the reviewers for their comments about our work, and the constructive questions and suggestions. We have performed additional experiments and theoretical analysis. We hope that our manuscript is now suitable for acceptance.

The referee reports brought out five main points of discussion. We give a detailed discussion to each of these points below. After that, we highlight point-by-point responses to specific referee comments.

The points of discussion of this letter are:

1) Testing temporal perturbation using a gap gene network model

2) The mechanism underlying light-induced Bcd-dependent transcriptional inhibition

3) The kinetics of light-responsive loss and recovery of transcription activity

4) Measurement precision of gene expression profiles

4.1) Quantification of Cad gradients in altered Bcd dosages

4.2) Further support for “5% anterior shift” in embryos subjected to early illumination

5) Incorporating current findings with previous studies on Bcd-dependent patterning

5.1) Reconciling temporal Bcd interpretation with previous quantitative work

5.2) Insights from phenotypic resemblance between hypomorphic *bcd* alleles and short temporal perturbations

1) Testing temporal perturbation using a gap gene network model

Modelling has been a powerful tool in understanding early *Drosophila* embryo development, with models identifying critical components in the gap gene network required for forming robust gene expression boundaries (Bieler et al., 2011; Jaeger et al., 2004; Manu et al., 2009a, 2009b; Perkins et al., 2006). These models use coupled differential equations to incorporate interactions between gap genes (Hb, Gt, Kr and Kni) and with maternal inputs, along with diffusion and protein degradation.

To test how such models perform under temporal perturbation of the Bcd input, we used the model outlined in Bieler et al., which is an advanced gap gene network model that incorporates the three-dimensional embryo shape, regulatory inputs from Bcd, Cad, Tailless and Huckebein, and the distinct regulatory functions of Hunchback monomers and dimers. The general form of this model is similar to other published models (though Bieler et al., represents one of the most up-to-date versions), and therefore we do not expect significant variation in our conclusions from using other, similar, models.

We were able to effectively introduce into the model Bcd temporal deprivation by setting the interaction strength with Bcd to zero at different times in the simulation, see Materials and methods and new Figure 5—figure supplement 3. To briefly summarize these results, we find that our optogenetic perturbations enable us to refine the parameter estimations for the interaction parameters between gap genes and between gap genes and maternal inputs. Bieler et al., identified 21 in silicoparameter sets that agree well with observed gap gene expression profiles. Of these, a subset of 9 were qualitatively consistent with our observed optogenetic perturbations. We find that the parameter values representing regulation of Hb and Kr in this subset do not show much variation compared with the average parameter values for all 21 parameter sets. However, the parameters regulating Gt and Kni expression show larger variation between the subset and the complete parameter set. For example, on average the input of Bcd into Gt regulation is increased and Kni autoregulation is decreased in the parameter subset.

Results of these numerical simulations are provided in the main paper and in Figure 5—figure supplement 3, with a new Materials and methods section outlining details of how the simulations were performed (subsection “Refining parameter estimation in models of gap gene network”. The simulations provide two key insights: (1) they enable refinement of existing parameter sets, in particular by incorporating a more dynamic perturbation; (2) they predict that additional inputs are required for the model, particularly in the anterior of the embryo, to match our observations.

2) The mechanism underlying light-induced Bcd-dependent transcriptional inhibition

At the early phase of the project when we first observed the light-induced lethality in embryos expressing maternally loaded CRY2::mCh::Bcd protein in a wild-type bcd background, we considered several possible mechanisms underlying such phenomenon. (1) Light induces CRY2 dimerization/oligomerization and therefore the formation of CRY2::mCh::Bcd protein clusters. Wild-type Bcd proteins are also recruited into clusters due to Bcd-Bcd protein interaction. The enormous size of such clusters prevent Bcd proteins from entering nuclei or freely diffusing therefore abolishes the activity of Bcd-dependent transcription. (2) The conformational change of CRY2 masks the surface of Bcd protein that is essential for protein-protein interaction. This perturbs the cooperative binding of Bcd to its target sites and therefore alters the gene expression patterns. (3) The CRY2 conformational change affects the proper folding of the homeodomain, resulting in the failure of target recognition or ectopic DNA binding in the genome. (4) The change of the N-terminal CRY2 structure causes a conformational hindrance in the assembly of the transcription machinery, thus abolishing target gene expression.

Several lines of experimental evidences helped us to narrow down the possible scenarios. First, we carried out live imaging of CRY2::mCh::Bcd embryos and utilized the mCherry signal to indicate clustering events. In both dark and illuminated conditions, no obvious clusters were observed in the cytoplasm or in nuclei, even at the anterior-most region of the embryo where Bcd is present at its highest concentration (Figure 7). Neither was the Bcd gradient profile significantly altered indicating a normal diffusion rate. Second, both cuticle patterns and segmentation gene expression patterns of illuminated embryos show more severe phenotypes compared to Bcd cooperativity mutants (Lebrecht et al., 2005), although we do not rule out the fact that cooperative binding between Bcd proteins could also be interfered by light response. Third, the fact that CRY2::mCh::Bcd inhibits Caudal translation in the anterior region of illuminated embryos suggests that CRY2::mCh::Bcd in light can specifically recognize and bind to the BRE (Bcd response element) in the 3’UTR of Caudal mRNA. This indicates that the homeodomain folding remains intact under illuminated condition. Fourth, CRY2::mCh::Bcd competes with wild-type Bcd to inhibit target gene transcription in a dominant negative manner, indicating that CRY2::mCh::Bcd in light still binds to most of its native DNA binding sites. Finally, a cellularization phenotype was previously characterized in *bcd* mutant embryos (excessive membrane ingression in the anterior end of the embryos) although the linking factors between Bcd morphogen to the regulation of cytoskeletal components remains unclear (Blankenship and Wieschaus, 2001). We found that light also induces such cellularization phenotype in CRY2::mCh::Bcd embryos (Figure 7), again indicating that CRY2::mCh::Bcd in light still binds to its native target sites faithfully.

Author response image 1.Exploring the mechanism underlying optogenetic manipulation of Bcd-dependent transcription.(**A**) Live imaging of cry2::mch::bcd, *bcd*^-/-^ embryo in dark (left panels) and illuminated (right panels) conditions. The top panels show the zoom-in of most anterior embryo regions. (B – D) Cross-section of late cycle 14 embryos stained for Phalloidin. Cry2::mch::bcd, *bcd*^-/-^ embryo in dark (**B**) and illuminated (**C**) conditions are compared to *bcd* null embryo (**D**). Red bars indicate the depth of membrane ingression in the anterior end of the embryos (shallow cellularization front in dark embryos versus excessive ingression in illuminated and *bcd^-/-^* embryos).**DOI:**
http://dx.doi.org/10.7554/eLife.26258.020

Altogether, these results point to scenario (4), where light-induced CRY2 conformational change does not alter Bcd specific DNA binding but abolishes its transcription activating potency. This renders the construct ideal for studying temporal morphogen interpretation. The difficulty here lies in specifying the molecular mechanism underlying such inhibitory effect as well as ruling out any unspecific binding effects. Roles of Bcd N-terminal domain in mediating protein interactions were elucidated previously (Fu et al., 2003; Ma et al., 1996). Here it is unclear whether it is Bcd-Bcd interaction, Bcd interaction with its co-activators and/or other general transcription cofactors that is inhibited. Perhaps future structural studies on Cryptochromes light-responsive conformational changes will help us to clarify this question.

We have included the above points in the Discussion, subsection “Mechanism underlying light-induced inhibition of Bcd transcription activity”.

3) Kinetics of light-responsive loss and recovery of transcription activity

Previously, we have used the MS2 coat protein (MCP) tagged with GFP to detect nascent transcripts driven by the *hb* P2 promoter. The limitation of this reporter system lies in that we cannot visualize the transcription under dark condition in CRY2::mCh::Bcd embryos as the GFP excitation wavelength (488 nm) overlaps with CRY2 light-sensitive range (405–488 nm). We have recently developed and characterized a new reporter system where the MCP is tagged with mCherry instead. The updated findings are (1) MCP::mCh recapitulates the hb transcription dynamics observed with MCP::GFP; (2) Illumination (by LED light source) caused instantaneous and complete shutdown of hb transcription; and (3) The transcription reinitiates 5 minutes after cessation of illumination. These results are quantitatively consistent with CRY2 light-responsive kinetics reported previously (Kennedy et al., 2010). These results are described in the fourth paragraph of the subsection “Precise temporal control of Bcd-dependent transcription via optogenetic manipulation” and Figure 2—figure supplement 3 and Figure 2—figure supplement 4.

4) Precision in measurement of gene expression profiles

Gap gene profiles display extreme reproducibility in *Drosophila* embryos given well-defined stages (Gregor et al., 2007; Houchmandzadeh et al., 2002). We have made similar measurements on control embryos (OreR) and the variation of most gap gene boundary positions is within the range of 1~2% embryonic length (Figure 8). Therefore, we are confident that we can detect real (though small) shifts in the gene expression boundaries.

Author response image 2.Precise measurements on four gap genes.(A – D) Max projection images of *Drosophila* embryos stained for Hb (A), Kr (B), Kni (C) and Gt (D) fixed at around 40 min into cycle 14 (top panels). Average intensity normalized to peak value is plotted vs. AP position (% EL) for each gap gene (bottom panels). Shaded error bars are across all nuclei of all embryos at a given position. n = 7-9 embryos for each gap gene quantification.**DOI:**
http://dx.doi.org/10.7554/eLife.26258.021

4.1) Quantification of Cad gradients in altered Bcd dosages

To verify our observation that the Cad gradient is not altered by the optogenetic perturbation of Bcd transcriptional activity, we tested the sensitivity of our assay by measuring the Cad gradient in embryos with one, two or four maternal copies of bcd. As shown in the new Figure 1—figure supplement 1, we saw gradual posterior shift of Cad boundaries with increased bcd dosage. We quantified the Cad boundary position at which each embryo profile crosses the half maximum posterior expression value, and found that boundary positions in altered bcd dosages are significantly different (51.32 ± 4.95% EL, 45.55 ± 5.05% EL and 38.91 ± 3.66% EL for 1x, 2x and 4x bcd, respectively).

4.2) Further support for “5% anterior shift” in embryos subjected to early illumination

We have noticed that embryos illuminated before n.c.12 show a 5% anteriorly shifted Hb posterior boundary in the anterior domain as well as Kr anterior boundary, although these embryos are highly viable. A recent ChIP-seq study has shown that Bcd molecules serve a role of opening up chromatin states at a subset of its targets in early stages to gain accessibility for transcription (Hannon et al., 2017). Our data suggests that early Bcd-dependent transcription events may have a chromosomal remodeling effect for the target sites to interpret a normal Bcd concentration in later stages. If embryos are deprived of its early Bcd-dependent transcription, target sites ‘see’ a lower Bcd gradient due to the compromised chromatin accessibility. To further support this observation, we have quantified the position of the cephalic furrow which shifts its position along the AP axis with altered bcd dosage. In consistency with our postulate, we observed a significant anterior shift of cephalic furrow position in embryos illumination before n.c.12 compared to dark condition (64.4 ± 1.5% EL in dark and 70.4 ± 1.3% EL in illuminated embryos; new Figure 5—figure supplement 2 and subsection “Temporal dissection of Bcd target gene expression”, second paragraph).

5) Incorporating current findings with previous studies on Bcd-dependent patterning

5.1) Reconciling temporal Bcd interpretation with previous quantitative work

Studies from Gregor’s lab have been focusing on using information theory to understand the precision and reproducibility in Bcd-mediated embryonic patterning. Among many important findings, the most relevant to the current study are (1) While early target gene boundaries are set by the absolute Bcd concentration, the cross-regulatory gap gene network kicks in later to adjust the boundary positions and therefore Bcd is *no longer relevant* for patterning precision (Liu et al., 2013); (2) The embryonic patterning operates in a precisionist mode in that the information carried by gap genes at one defined time point is not only sufficient but also fully utilized by cells for *decoding positional information*, as probed by pair-rule genes (Dubuis et al., 2013; Petkova et al., 2016).

Our findings do not dispute the important role of the gap gene network in refining segmentation boundaries. In fact, embryos subjected to short period of illumination in late n.c.14 show only slight gap gene boundary shifts as a result of self-organized gap gene cross-regulation. However, we argue against the claim that Bcd in late n.c. 14 becomes irrelevant for patterning for the reason that, as we have shown, Bcd-dependent transcription is indispensable at this time for sustaining the proper expression level of gap genes, especially the most anteriorly expressed ones (which are not considered in Liu et al., 2013). The proper expression level of the anterior gap genes is in turn essential for precise anterior-most cell fate commitment.

Our results are in agreement with a potentially optimized information flow from gap genes to downstream pair-rule gene patterns. We anticipate that the gap gene expression patterns under different illumination conditions can predict the pair-rule gene patterns using the same decoding algorithms as established by Petkova et al., 2016. However, what is not yet considered in these studies is the information flow from a snapshot patterning to the actual embryonic cell fate – just like Bcd, all the gap genes as well as pair-rule genes are transcription factors of which the duration of activity matters to cell fate determination. We have shown that indeed cell fates are determined not only by segmentation gene expression at a given time, but also its integration over time.

5.2) Insights from phenotypic resemblance between hypomorphic bcd alleles and short temporal perturbations

Three anterior embryonic regions with different sensitivity to bcd^+^ reduction can be distinguished, that is, the preantennal head region, the antennal and gnathocephalic head segments, and the thorax. While the weakest *bcd* alleles only affect the first region, defects sequentially extend to the second and third regions with increased allelic strength (Frohnhofer and Nusslein-Volhard, 1987). Interestingly, the same distinction can be made by our temporally perturbed Bcd activity, with prolonged transcriptional inhibition giving rise to defects in sequentially extended regions.

The molecular properties of these hypomorphic *bcd* alleles affect either the DNA binding affinity (e.g. *bcd*^E3^, point mutation in homeodomain) or the activation potency (e.g. *bcd*^2-13^, partial deletion of the C-terminal activation domain) of the Bcd protein. The resemblance between hypomorphic *bcd* alleles and short temporal perturbations demonstrates that, constitutively low bcd activity is functionally equivalent to normal bcd activity with reduced active duration, both of which are sufficient to guide proper cell differentiation trajectories in the mid region of the embryo where target gene binding affinities are high. In comparison, in the more anterior region where target genes have low affinities, both wild-type level bcd activity and long duration are necessary for anterior induction (Figure 9). It is noteworthy that duplicated posterior structures only appear in strong allelic mutants but not in embryos subjected to long illumination due to a normal Caudal gradient in our experimental condition. It would be interesting to understand whether hypomorphic *bcd* alleles can suppress Caudal translation.

We have included within the Discussion a more detailed comparison between our optogenetic perturbations and the previously reported different *bcd* alleles (subsection “Phenotypic resemblance between hypomorphic bcd alleles and short temporal perturbations”).

Author response image 3.Embryonic regions distinguished by different sensitivity of Bcd activity and duration.Red, orange and blue segments represent distinct embryonic regions requiring Bcd activation potency at WT, intermediate hypomorphic and weak hypomorphic level for proper differentiation, respectively. Concomitantly, they require Bcd for long, intermediate and short duration in time.**DOI:**
http://dx.doi.org/10.7554/eLife.26258.022

*Reviewer #1:*

*For the last 30 years, the Bicoid gradient has been the most studied example of a morphogenetic gradient. It would thus seem difficult to find new important details that have been missed by these studies that went deep into molecular mechanisms and models of gene regulation. Yet, this paper appears to provide just that: not a huge amount of conceptual novelty, but a powerful way to manipulate temporal parameters of the gradient using a very clever method to render (some of) the activity of the protein light dependent.*

*The authors achieved this by fusing the Bicoid protein in the context of its own rescue construct to a light sensitive moiety. Interestingly, this appears to only affect the transcriptional activity of the protein and not its DNA binding or its RNA regulatory capacity: the activation domain might be more sensitive, which is surprising but the data show a full rescue without light and no rescue with light. This allowed the authors to stop activation by Bicoid at various times during development. They conclude that the temporal pattern of Bcd activation is important, which appears to be in slight contradiction with the work of Thomas Gregor who also quantified very carefully the effect of Bicoid but could not manipulate it so precisely. It appears that genes that require higher Bcd concentration require Bcd for longer duration, while exposed to low Bcd (whose targets have high affinity Bicoid binding sites) can be specified with short early exposure.*

Please see section 5.1above, where we discuss our results in relation to other labs, particularly those of Thomas Gregor.

*Reviewer #2:*

*[…] The authors characterize the CRY2::mCh::Bcd in Figure 1 and 2. Additional information would be helpful and some insight into the mechanism would be informative. This seems particularly important, given the dominant negative activity of the construct.*

We discuss possible mechanisms in the above section 2. We have also incorporated the main points into the Discussion, subsection “Mechanism underlying light-induced inhibition of Bcd transcription activity”.

*The authors suggest that illuminated embryos "resemble" the bcd knockout phenotype (subsection “Precise temporal control of Bcd-dependent transcription via optogenetic manipulation”, second paragraph). In Figure 1—figure supplement 2 the authors show that the phenotypes are not identical. This should be highlighted earlier in the text and the phenotypic comparison given more interpretation, to help readers not expert in the system.*

In the second paragraph of the subsection “Precise temporal control of Bcd-dependent transcription via optogenetic manipulation” we compare the phenotypes of *bcd* null and illuminated bcd^CRY2^ embryos in detail and explained the cause of such phenotypic differences.

*The mechanism by which the cryptochrome cassette blocks Bcd activity is unclear. The authors note that spatial distribution of Bcd and DNA binding activity is unaffected and the Caudal gradient is also unaltered. These experiments would be strengthened by controls in which Bcd and Cad gradients are disrupted, to gauge the sensitivity of the assays. The genotypes of these embryos should also be clearly stated in the figure legend.*

We have tested the sensitivity of the assays by measuring Cad gradients in embryos with altered Bcd dosages (1, 2, and 4 copies of maternal bcd), and found significant shifts in Cad boundary positions (new Figure 1—figure supplement 1). The measurement is also sensitive enough to see the difference of Bcd profiles in embryos maternally expressing Bcd at different levels due to different insertion loci (Figure 2—figure supplement 1), supporting our argument that Bcd and Cad distributions are not changed in illuminated conditions compared to dark. We have stated the genotypes in the legends.

*Experiments that directly test how quickly Bcd dependent transcription is lost in illuminated conditions and recovered following cessation of illumination would strengthen the interpretation of Figure 3.*

We agree that such experiments are helpful. To this end, we have tagged the MCP protein with mCherry, enabling us to observe the system without triggering CRY2 conformational changes (which was an issue with the MCP::GFP protein). Full details are provided above in section 3. In the manuscript new supplemental figures are included – Figure 2—figure supplement 3 and Figure 2—figure supplement 4 – along with detailed description in the fourth paragraph of the subsection “Precise temporal control of Bcd-dependent transcription via optogenetic manipulation”.

*Figure 4 demonstrates the sequential loss of identities from the anterior as the illumination times are increased. It would be helpful to have a schematic of gene expression in this figure. While the loss of anterior gene expression is evident, the anterior expansion of more posterior fates (as would be expected in a homeotic-like transformation) is less evident.*

In new Figure 4 we have added schematics of Hox gene expression profiles. We believe this makes the interpretation of the figure easier.

*The data in Figure 5 and Figure 6 provide a detailed quantitative description of the temporal requirements for Bcd activity. These are novel and interesting findings that will help move the Bcd patterning field forward. Given the authors' expertise and the availability of dynamical models of the Gap gene system (from Reinitz, Jaeger and others) it seems a missed opportunity not to test whether in silico simulations using the existing models (interrupting Bcd activity during defined time windows) recapitulates the experimental results. Although developing an entirely new mathematical model might be beyond the scope of the current study, testing whether the data fit existing models would indicate whether new models (or new parameters for existing models) will be necessary.*

We agree that testing our results against current models of gap gene systems is beneficial. We have performed such an in silicoexperiment using the model of Bieler et al., 2011. A complete discussion is given above in section 1. We have included in the Results of the manuscript (subsection “Refining parameter estimation in models of gap gene network”) a discussion of the *in silico* results and a new Figure 5—figure supplement 3.

*Related to this, the authors might want to extend their Discussion to consider the involvement of the dynamics of the Bcd controlled transcriptional network (as well as the kinetics of individual gene responses) in the differential sensitivity to durations of Bcd activity. This would allow a more direct comparison with the vertebrate neural tube where the dynamics of the Shh responsive transcriptional network are implicated.*

We discuss the similarities with the vertebrate neural tube in the Discussion (subsection “The temporal pattern of Bcd signal integration is spatially inhomogeneous”, last paragraph). We fully agree that confronting our results with that of the Shh transcriptional network would be insightful. However, it would require extensive work and refinement of the current gap gene models.

*Reviewer #3:*

*Huang et al. describe an optogenetic manipulation approach for investigating the temporal requirement of the Drosophila morphogen Bicoid (Bcd) for embryonic development. Their general conclusion is that cell fates dependent on higher Bcd concentration require the Bcd input for a longer duration of time. The concepts are interesting and most of the experiments are well controlled. The findings reported here are useful to the field. There are a few aspects that can be improved.*

*While the observed effects of light exposure on embryonic development and gene expression are convincing, the underlying mechanisms remain relatively unclear. This is because precisely how the CRY2::mCh::Bcd fusion protein becomes inactivated at the molecular or biochemical level has not been established. The authors describe experiments suggesting that neither the ability of Bcd in translation inhibition (monitored by Cad protein gradient profile) nor its DNA binding (monitored by ChIP) was affected. It is possible that, as suggested by the authors, the N-terminal fusion of CYR2 to Bcd may cause a light-induced conformational change that inhibits the transcriptional activity of Bcd (since its N-terminus does contain a regulatory motif modulating its transcription activity). But I am not fully convinced that the ChIP approach can sensitively and unambiguously rule out an effect on DNA binding. The fact that the expression domains of hb and Kr exhibit an anterior shift (Figure 5) is consistent with a potential effect on Bcd DNA binding (or Bcd gradient profile might also be altered – Figure 1). While the distinction between DNA binding and transcriptional activation is crucial to understanding Bcd action as a transcription factor and several reports have recently described perturbations specifically altering the activation potency of Bcd, I believe the current manuscript can be written in a way that sidesteps this issue; but if the authors prefer to discuss it, they may need to provide a more detailed Discussion. Interestingly altering the activating potency of Bcd is known to cause changes in the shutdown timing of hb transcription and the hb expression level at nc14 in a specific manner. Whether and, if so, how these results are related to the current findings (particularly with regard to the persistence of transcription at nc13) should be discussed to give the reader a more comprehensive update of the current knowledge.*

We agree that ChIP-qPCR experiment does not have the resolution to rule out any unspecific DNA binding caused by CRY2 conformational change. However, several experimental observations support that light-induced conformational change does not alter DNA binding of CRY2::mCh::Bcd molecules. We provide a detailed discussion of possible mechanisms and evidence for each one in section 2above. We have also extended the Discussion (subsection “Mechanism underlying light-induced inhibition of Bcd transcription activity”) to outline the mechanism underlying Bcd-dependent transcriptional inhibition in our experiments.

Moreover, embryos illuminated before n.c. 12 should have wild-type Bcd activity present during n.c. 13-14; however, we still observed anterior shifts of Hb and Kr in these embryos. Since we have confirmed that the Bcd profile is not altered by illumination, we attribute the anterior shifts to the pre-patterning role of Bcd in early stages (see above section 4.2).

Lastly, while local illumination by confocal microscopy results in a reduced persistence of transcription in n.c. 13, a thorough illumination by LED source can immediately and completely abolish transcription in n.c. 13 (as shown in Figure 2—figure supplement 4).

*It is generally acknowledged that weaker bcd alleles tend to affect preferentially the formation of the most anterior structures, suggesting that such structures are more sensitive to perturbations of the Bcd input. It would be useful to discuss how this study may (or may not) provide mechanistic insights into those earlier findings.*

We discuss these points in detail above in section 5.2. In the paper, we have extended the Discussion (subsection “Phenotypic resemblance between hypomorphic bcd alleles and short temporal perturbations”) to incorporate key points related to this comparison.

*It is useful to provide a brief discussion on the estimated time scales of both illumination-induced inactivation and "reactivation" in experimental setting described in the current study.*

We agree, and with our MCP::mCherry construct we are now able to image the developing embryo without simultaneously triggering CRY2 conformational changes (which was a problem with MCP::GFP). Full details are provided above in section 2. In the manuscript new supplementary figures are included – Figure 2—figure supplement 3 and Figure 2—figure supplement 4 – along with description in the fourth paragraph of the subsection “Precise temporal control of Bcd-dependent transcription via optogenetic manipulation”.